# Low level of plasma DNase is associated with worse clinical outcome in testicular germ cell tumor patients and exogeneous DNase I improves cisplatin treatment efficacy

**Michal Mego** [1,2,3]*, **Barbora Vlkova**[4], **Katarina Kalavska**[2,3], **Michal Pastorek**[4], **Zuzana Cierna**[5,6,7], **Zuzana Sestakova**[3], **Miroslav Chovanec**[3], **Natalia Udvorkova**[3], **Lucia Kucerova**[2,3‡], **Peter Celec**[4,8‡]

**1** 2nd Oncology Clinic, Faculty of Medicine, Comenius University and National Cancer Institute, Bratislava, Slovakia, **2** Translational Research Unit, Faculty of Medicine, Comenius University and National Cancer Institute, Bratislava, Slovakia, **3** Cancer Research Institute, Biomedical Research Center, Slovak Academy of Sciences, Bratislava, Slovakia, **4** Institute of Molecular Biomedicine, Faculty of Medicine, Comenius University, Bratislava, Slovakia, **5** Department of Pathology, Faculty of Medicine, Comenius University, Bratislava, Slovakia, **6** Department of Pathology, Faculty Hospital, Trnava, Slovakia, **7** Faculty of Health Care and Social Work, Trnava University in Trnava, Trnava, Slovakia, **8** Institute of Pathophysiology, Faculty of Medicine, Comenius University, Bratislava, Slovakia

‡ These authors shared last authorship on this work.
* misomego@gmail.com

## Abstract

### Background

Germ cell tumor (GCT) patients with unfavourable response to first-line therapy still lack reliable diagnostic and effective treatment Detailed correlation of total extracellular DNA (ecDNA), other DNA species and endogenous DNase levels in GCT patients' plasma and translational utility remains under-investigated.

### Study aim and methods

We determined DNase plasma levels, ecDNA of different subcellular origin and neutrophil extracellular trap (NETs)-associated markers. Next, we determined the associations of these parameters with a level of the DNA damage, immune inflammatory index, specific immune cell subpopulations in a cohort of the 117 GCT patients and 19 matched healthy donors (HDs). Moreover, we investigated how exogenous DNase affects antitumor effect of cisplatin in GCT model of cisplatin-resistant embryonal carcinoma NTERA-2 CisR.

### Results

Our data demonstrate that high level of ecDNA and low level of DNase in GCT patients' plasma is associated with significantly worse progression-free survival and overall survival. The level of the plasma ecDNA was five times higher in the GCT

**Data availability statement:** Data contain potentially identifying or sensitive patient information according to Institutional Ethics Committee. Contact for a data access committee: Ethics committee of National Cancer Institute, contact person: RNDr.Daniela Svetlovska, PhD, phone +421 2 59378 592, www.nou.sk).

**Funding:** This work was supported by the VEGA Grant Agency of the Slovak Republic (VEGA 1/0349/21, 2/0124/21 and 2/0075/23) and the Slovak Research and Development Agency (grant APVV-20-0158 and APVV-22-0554). The sponsors played no direct role in the study. This work was also supported by the Integrated infrastructure operational program for the projects 'Systemic public research infrastructure - Biobank for cancer and rare diseases' (ITMS grant number 313011AFG5) co-financed by the European Regional Development Fund.

**Competing interests:** The authors have declared that no competing interests exist.

**Abbreviations:** bpDNase, bovine pancreatic; cDCs, conventional dendritic cells; CDDP, cisplatin; CR, complete remission; DC, dendritic cell; EC, embryonal carcinoma; ecDNA, extracellular DNA; FSC, forward scatter; GCTs, germ cell tumors; GE, genome equivalents; IGCCCG, international germ cell collaborative group; iMVD, intratumoral microvascular density; L, lymphocytes; MMP9, matrix metalloproteinase 9; MPO, myeloperoxidase; mtDNA, mitochondrial DNA; N, neutrophils; ncDNA, nuclear; NE, neutrophil elastase; NETs, neutrophil extracellular traps; OS, overall survival; P, platelets; pDCs, plasmocytoid dendritic cells; PFS, progression-free survival; PNMs, polymorphonuclear monocytes; rhDNase, recombinant human DNase; SD, standard deviation; SEM, standard error of the mean; SII, systemic inflammatory index; T/TC, teratocarcinoma.

patients compared to the HDs. The patients with higher total ecDNA and ncDNA, but not mtDNA, had inferior PFS and OS compared to the patients with lower ecDNA (all $p < 0.05$). There was an inverse correlation between plasma DNase and ecDNA levels, and between plasma DNase level and clinical outcome. Importantly, combined treatment with cisplatin and human recombinant DNase I delayed growth of the NTERA-2 CisR xenografts and prolonged animal survival. Importantly, Pulmozyme significantly reduced intratumoral microvascular density in our preclinical model.

## Conclusion

Our data confirm the association between low plasma DNase activity and worse overall survival for the first time in GCT patients. This study further validated the prognostic value of total ecDNA in GCT patients. More importantly, our preclinical data substantiated beneficial effect of Pulmozyme combination with cisplatin treatment to improve the therapeutic outcome in refractory disease.

## Introduction

Germ cell tumours (GCTs) are extraordinarily chemosensitive and resemble the clinical and biological characteristics of a model for the cancer cure [1]. Nonetheless, a small proportion of patients do not achieve a durable complete remission (CR) with initial chemotherapy. Only 20–40% of them can be cured with the use of platinum-containing standard-dose or high-dose salvage chemotherapy with autologous stem cell transplantation [1]. Patients who fail to be cured after second-line salvage therapy have an extremely poor prognosis and long-term survival is documented in less than 5% [2–5]. Paclitaxel plus ifosfamide and cisplatin (CDDP) is considered as a standard salvage chemotherapy in relapsed good prognosis GCTs. However, up to 40% of favorable prognosis patients failed to achieve durable response with this combination, and therefore new treatment strategies are warranted [4,6–9].

Tumor-associated cell-free or extracellular DNA (ecDNA) is applicable as a biomarker in screening and monitoring at different stages of tumor progression [10–12]. Interestingly, total ecDNA in plasma has a prognostic value as well [13,14]. The dynamics of total ecDNA in cancer patients and its association with the prognosis raised also questions about the role of ecDNA in tumor progression and metastasis [15–17]. Besides the tumor, healthy tissues and immune cells contribute to the pool of ecDNA in plasma with the nuclear and mitochondrial genomes as the subcellular source [18,19]. An active release of DNA was demonstrated during formation of neutrophil extracellular traps (NETs) [20]. As a part of the innate immune defense, DNA is released from neutrophils together with histones, neutrophil enzymes, antimicrobial peptides and other NETs components to trap and eradicate pathogens [21]. Clinical and experimental studies highlight the pivotal role of neutrophils and their NETs in inflammation, thrombosis, and cancer [22–25]. NETs were found in liquid and tissue biopsies of cancer patients. Over the last years, many studies linked the formation of

NETs to oncogenic transformation, angiogenesis, cancer development and metastasis, and unraveled their role in influencing responses to anticancer therapies [26,27].

Tumor cells can directly induce the formation of NETs *via* various mechanisms [20,28]. On the other hand, several NET components can induce tumor growth and migration [29]. DNA serves as a scaffold and trapping component but may act *via* TLR9 or CCDC25 receptor binding. HMGB1, a DNA-binding protein, and components of neutrophil granules, namely neutrophil elastase (NE) and reactive oxygen species, activate tumor cells. NE and matrix metalloproteinase 9 (MMP-9) cut laminin, which induces cascades that result in tumor cell proliferation [28,30,31]. Other neutrophil weapons such as cathepsin G or proteinase 3 are also apparently capable of activating tumor cells. In addition, neutrophils can activate platelets through P-selectin glycoprotein ligand-1 – P-selectin interactions with TLR4, leading to cancer-associated thrombosis. [28].

The main DNA-hydrolyzing enzyme in the blood is DNase I belonging to the DNase I family [32], and changes in the activity of DNases in the course of tumorigenesis were reported [28]. In the early phase, intracellular DNases and later DNase I participate in tumor cell apoptosis by destroying damaged DNA. At the same time, cells of the tumor environment produce and secrete DNase I into the intercellular space, which also promotes apoptosis. If apoptosis fails, transformed cells accumulate DNases, which are either present in an inactive state in the cytoplasm or accumulated on the surface of the cell [28]. Surface nucleases protect tumor cells from exogenous DNA and DNA from NETs, thereby preventing the catch and destruction of tumor cells during the early stages of tumorigenesis [33]. During cell transformation, DNase I is inactivated by binding with actin filaments (mostly G-actin) and anti-DNase antibodies [28]. These changes lead to a decrease in DNase activity in the tumor microenvironment that reduces the apoptotic pressure on tumor cells. In addition, a low DNase activity leads to the accumulation of NETs; induction of inflammation; this all leads to the detachment, migration and metastasis of tumor cells [28].

Total DNase activity in plasma has been found to differ between healthy donors and cancer patients. The earliest data showed that patients with malignant lymphomas were characterized by lower DNase activity, whereas patients with breast cancer demonstrated higher levels of DNase activity in comparison with healthy donors [34]. Patients with oesophageal cancer or prostate cancer are also characterized by very low DNase activity, while ecDNA concentrations are significantly higher [35,36]. For patients with liver cancer, a correlation has been found between a high DNase activity and low concentration of ecDNA with the risk of developing cancer in the preclinical setting. Nevertheless, tumor progression in patients resulted in a decrease in the activity of DNases, which is accompanied by an increase in the level of ecDNA [37]. Importantly, study by Hernandez *et al.* suggested that restoration of NETs degradation *in vitro* can be achieved by DNase I treatment and suggested that the recombinant human DNase I Pulmozyme® could locally reduce bladder cancer progression [38].

Deoxyribonuclease (DNase) enzymes degrade ecDNA to prevent immune system activation [32]. Cancer studies have demonstrated that DNase has the capacity to modify tumor microenvironment to enhance the penetration of chemotherapy into the tumor [28]. Moreover, preclinical research has unveiled potential of DNase to modulate the immune response within the tumor. DNase could facilitate the infiltration of immune cells into tumors, thereby potentially augmenting the efficacy of immunotherapeutic approaches [28].

DNase has also been shown to interfere with the metastatic process. In preclinical models, DNase disrupts NETs thereby curbing the propagation of cancer cells to distant sites and restricting the metastatic spread [28,39,40]. Additionally, the anti-inflammatory effects of DNase have been observed in preclinical studies [41]. As inflammation plays an important role in tumor progression, mitigating inflammatory responses through DNase treatment could potentially slow down tumor growth [28,41]. Combining DNase treatment with other therapeutic modalities has emerged as an exciting avenue in preclinical investigations. The potential synergistic effects of DNase with chemotherapeutic agents or immunotherapies show promise in enhancing overall treatment efficacy by bolstering drug delivery, anti-tumor immune reactions, and dampening tumor-related inflammation [39,42].

In a preclinical B16 mouse melanoma model, the recombinant human DNase I was repurposed for the inhibition of lung metastasis formation. The treatment strongly reduced migration and induced apoptosis of melanoma cells B16 *in vitro* and effectively inhibited metastases in lungs and liver *in vivo.* DNase I significantly decreased ecDNA concentrations and induced apoptosis and disintegration of NETs in metastatic foci. This manifested as inhibition of metastases spread. Based on these results, DNase I was suggested for the treatment of lung metastases [43]. Another trial demonstrated that systemic treatment with DNase I and a protease mix in rats decreases ecDNA and has antitumor effects [44]. Study by Mousset *et al*. suggested the use of DNase I for the improvement of efficacy of CDDP and adriamycin/cyclophosphamide treatments in an experimental model of breast cancer metastasis [39].

In this study, we have focused on the association of plasma ecDNA and DNase activity with patient/tumor characteristics and the outcome in GCT cohort. We employed multiple *in vitro* tests and *in vivo* models to evaluate the potential of the DNase I in augmentation of platinum-based therapy in refractory/recurrent GCT patients.

## Patients and methods

**Study patients.** This translational study (Protocol IZLO1. Chair: M. Mego) included GCTs patients who underwent treatment with systemic chemotherapy from 1st October 2012 to 31st January 2020 and for whom plasma isolated on the day 1, before systemic chemotherapy, was available in the biobank. All patients with GCTs treated with at least one cycle of chemotherapy in the National Cancer Institute or St. Elisabeth Cancer Institute were enrolled in this prospective study. Plasma samples from age-matched healthy men were used as a control group. Data regarding age, tumor histological subtype, clinical stage, type and the number of sites of metastasis, and type of chemotherapy regimen were recorded in all the patients. GCTs patients and controls were recruited and consented according to the Institutional Review Board approved protocol. From 02/05/2023 were data accessed for research purposes.

**Plasma samples collection.** Peripheral venous blood samples collected in heparin and EDTA-treated tubes were centrifuged at 1,000 g for 10 min at room temperature (RT) within 2 h of venipuncture and processed as described previously [45]. Plasma aliquots were stored at −80ºC until further analysis. Blood was collected in the morning on the day 1 of adjuvant (n = 13), first line (n = 91) or salvage (n = 13) chemotherapy of the 1st cycle of chemotherapy (n = 117) and before the 2nd cycle of chemotherapy (n = 50).

**Analysis of plasma and microparticle-associated ecDNA.** Pre-treatment concentrations of the total plasma ecDNA, nuclear (ncDNA) and mitochondrial (mtDNA) ecDNAs were quantified in plasma using fluorometry and real-time PCR. Total ecDNA was isolated from double centrifuged EDTA blood plasma (1,600 and 16,000 g for 10 min at 4°C) using the QIAamp DNA Mini Kit (Qiagen, Hilden, Germany). The pellet after the second centrifugation was collected and used for the microparticle-associated DNA analysis as described before with following modifications [46]. Briefly, plasma was diluted 100 times and stained with 200 nM SYTOX Green™ for 30 min, put on ice and immediately analysed on DxFlex cytometer (Beckman) using a vertical side scatter on a 405 nm laser and 525/40 filter set on 488 nm laser according to Flow Cytometry Sub-micron Particle Size Reference Kit and Nonionic Latex Beads, 4% w/v, 5 µm (Thermo Fisher, Los Angeles, CA, USA). Gating strategy is showed in S1 Fig. The quantification of total ecDNA was conducted with Qubit dsDNA HS Assay Kit (Thermo Fisher, Los Angeles, CA, USA). In addition, the isolated ecDNA was used as a substrate for quantitative PCR targeting unique mitochondrial and nuclear sequences using a SybrGreen PCR mastermix (SsoAdvanced universal SYBR Green supermix, Biorad, Hercules, CA, USA). Primers used were as described previously for human and mouse ecDNA [47,48]. Efficiency of the real-time PCR reactions was between 90 and 110%. Melting curve analysis was conducted to prove specificity of the PCR products.

**Deoxyribonuclease activity.** Single-radial enzyme-diffusion assay was used for the assessment of DNase activity in heparin plasma as described before with modifications [49]. DNA-containing agarose gel [1% agarose gel consisting of 20 mM Tris·HCl (pH 7.5), 2 mM $MgCl_2$, and 2 mM $CaCl_2$, with DNA isolated from chicken livers – 0.35 mg/mL of gel] was used for the incubation of samples and the cleared circles in the gel were measured using ImageJ software (NIH,

Bethesda, MD, USA) and calculated as DNase activity in relation to a calibration curve with diluted DNase I from RNase-free DNase set (Qiagen, Hilden, Germany). Technical variability of the measurement was below 10%. As shown in S2 Fig. DNase activity but not ecDNA level decreased with increasing storage time. This could be a source of bias, but not for the survival analysis and not for ecDNA that seems to be stable over time.

**Determination of the DNA damage level in peripheral blood mononuclear cells (PBMCs).** The level of DNA damage in PBMCs isolated from GCT patients was determined using the comet assay as described previously [50,51].

**Systemic inflammatory index (SII).** The SII was determined using counts of peripheral blood platelets (P), neutrophils (N) and lymphocytes (L) *per* liter, which were retrieved from routine prechemotherapy blood tests. The equation $SII = P \times N/L$ was used as described previously [52,53].

**Determination of leukocyte immunophenotypes.** In the morning of day −1 or 0 of first line of chemotherapy, 1 mL atraumatic peripheral blood was collected into an EDTA-treated collection tube. Analyzed samples were processed within 24 h following collection, as previously described [50,51]. Briefly, leukocytes were stained using fluorochrome-conjugated antibodies (BD Pharmingen, USA) and, subsequently, leukocytes with defined immunophenotypes were quantified using flow cytometry (Canto II Cytometer; Becton, Dickinson and Company, Franklin Lakes, NJ, USA). The antibody combinations used for the basic panel, the regulatory T-cell panel, the dendritic-cell (DC) panel and the myeloid-derived suppressor-cell panel were used in the same scheme, as previously used [54]. A cocktail of used antibodies was incubated with 300,000–500,000 white blood cells in 200 µL for 20 min at RT. Before the fixation of cells using 1X BD FACS Lysing Solution (BD Bioscience, San Jose, CA, USA, cat. no: 349202), lysis of red blood cells was performed. For the assessment with a 1x BD FACSCanto™ II flow cytometer (Becton Dickinson, Franklin Lakes, NJ, USA), a minimum of 100,000 leukocytes were utilized. KALUZA software (Beckman Coulter, Inc., Brea, CA, USA) was used for the analysis of the flow cytometry data. Forward scatter (FSC) and side scatter were used to exclude debris according to size and granularity, while exclusion of doublets was performed using FSC-Height and FSC-Area. The number of gated cells considered as the minimum for evaluation was 100.

**Chemicals.** Following chemicals and substances were used: Deoxyribonuclease I from bovine pancreas (bpDNase, Protein ≥ 85%, ≥ 400 Kunitz units/mg protein, Sigma-Aldrich, Saint Louis, MO, USA), PULMOZYME® (dornase alpha – human recombinant DNase I, hrDNase; 1,000 U/1.0 mL, Roche Austria GmbH, Vienna, Austria), Cisplatin Accord 1 mg/mL (CPT, Accord Healthcare Polska Sp. z o.o., Poland), Puromycin solution (InvivoGen, Toulouse, France), TrypLE™ Express (GIBCO™), ECM Gel from Engelbreth-Holm-Swarm murine sarcoma (E1270, liquid, BioReagent, suitable for cell culture, Sigma-Aldrich®). Other chemicals were purchased from Sigma-Aldrich® (Saint Louis, MO, USA), if not stated otherwise.

**Cell lines.** This study employed established human GCT cell lines, namely embryonal carcinoma (EC) cell lines Tera-2, NTERA-2 and NCCIT, yolk sac tumor (YST) cell line NOY-1, teratocarcinoma (T/TC) cell line SuSa, and healthy human testicular fibroblasts Hs 1.Tes.

Cell lines NCCIT (ATCC® CRL2073™), NOY-1 (ENG101, Kerafast, Japan) and SuSa (ACC747, DSMZ, Braunschweig, Germany) were maintained in RPMI-1640 medium (Sigma-Aldrich®) containing 10% fetal bovine serum (FBS, GIBCO™), 10,000 IU/mL penicillin (Biotika, Slovakia), 5 µg/mL streptomycin and 1x GlutaMAX™-I Supplement (GIBCO™).

NTERA-2 (ATCC® CRL1973™) labeled NT2 cells throughout the manuscript, Tera-2 cells (ATCC® HTB-106™, kindly provided by Dr. Ludmila Boublikova, Charles University and University Hospital in Motol, Prague, the Czech Republic) and testicular fibroblasts Hs 1.Tes (ATCC® CRL-7002™) were cultivated in high-glucose (4.5 g/L) DMEM (Sigma-Aldrich®) supplemented with 10% FBS (GIBCO™), 10,000 IU/mL penicillin (Biotika, Slovakia), 5 µg/mL streptomycin and 1x GlutaMAX™-I Supplement (GIBCO™).

All cell lines were maintained in humidified atmosphere at 37°C under 5% $CO_2$. Cell line identities were confirmed by short tandem repeat (STR) profiling. Cell cultures were regularly screened to confirm their mycoplasma-negative status by MycoAlert™ PLUS Mycoplasma Detection Kit (Lonza, Köln, Germany).

The CDDP-resistant variants of NT2 and NOY-1 were generated by Dr. Katarína Kalavska and prof. Michal Mego, Translational Research Unit, Faculty of Medicine, Comenius University, Bratislava, Slovakia. Resistant isogenic variants were derived by a long-term propagation (≥ 6 months) of parental cells in sub-lethal concentrations of CDDP (CPT, Hospira UK Ltd., Queensway Royal Leamington Spa) without recovery time, as described previously [55]. Exposure started at 0.05 µg/mL CPT during exponential growth phase. When the cells started to expand, the CPT concentration was gradually increased to 0.1 µg/mL. Subsequently, *de novo* derived CDDP-resistant variants NT2 CisR and NOY-1 CisR were continuously maintained in 0.1 µg/mL CPT in culture media.

**Viability assays.** Quadruplicates of cultured cells were plated at $1\times10^3$ cells/100 µL media *per* well in white-walled 96-well plates (Greiner Bio-One GmbH, Kremsmünster, Austria). Compound(s) were diluted in respective culture media to reach designated final concentration and the cells were treated for 72 h. Relative endpoint viability was determined by the CellTiter-Glo® Luminescent Cell Viability Assay (Promega Corporation, Madison, WI, USA) and evaluated by the GloMax® Discover Microplate Reader (Promega). Value of relative viability units (RLU) of the untreated controls was taken as 100%. Experiments were performed at least three times, and the representative result is shown. Values were expressed as mean ± SD and $IC_{50}$ values were calculated using GraphPad Prism software (GraphPad Software, Boston, MA).

**Transduction of NT2 and NT2 CisR cells.** To generate cells with stable red fluorescent nuclear label, we transduced cells to express mKate2 protein ($\lambda_{abs}/\lambda_{em} = 588/633$ nm). Cells were seeded 24 h prior to transduction to reach 25–35% confluency at a time of infection. Incucyte® Nuclight Red (NLR) Lentivirus (puro) reagent (Cat. No. 4625, Essen BioScience Ltd. – A Sartorius Company, UK) was diluted in medium containing 8 µg/mL Polybrene® and added at MOI = 3/ cell. Cells were transduced at 37°C, 5% $CO_2$ for 24 h. Transduction medium was removed and refreshed for following 48 h culture. Cells were harvested, expanded and frozen. For stable expression, we performed antibiotic selection, adding 1 µg/mL puromycin to complete culture medium. Medium was replaced every 48–72 h and the expression of red fluorescent label was monitored in the Incucyte® Live-Cell Analysis System IncuCyte®ZOOM (Essen BioScience Ltd., Welwyn Garden City, UK). The chemosensitivity of transduced cells designated NLR-NT2 and NLR-NT2 CisR was determined by luminescent viability assay as described.

**Kinetic viability assay – live-cell imaging.** Quadruplicates of NLR labelled cells were plated at seeding density $1\times10^3$ cells/100 µL media *per* well in 96-well plates (TPP™ Cell Culture-Treated, Flat-Bottom Microplate). Compound(s) were diluted in respective culture medium to reach designated final concentration and added to the cells. Cell viability was monitored by the Incucyte®ZOOM system and images were taken with 10x objective in phase contrast to monitor culture confluence and red fluorescence channel every 2–3 h. Data were analyzed by IncuCyte®ZOOM software version 2016A. Data are expressed as mean ± SD of the Red Object (Cell) Count per Image.

**Migration and invasion assay.** NLR-NT2 CisR migration and invasion was evaluated by live-cell imaging according to the manufacturer's recommendation. Briefly, a selected area in 96-well plate (ImageLock 4397, IncuCyte® A Sartorius Brand, the US) was coated with 4% ECM diluted in medium (50 µL/well) for 30 min prior to cell seeding for invasion test. NLR NT-2 CisR cells were seeded at a density of 25,000 cells/well in 100 µL/well. The cells were allowed to settle at RT for 15 min, then incubated overnight prior to wounding. Wounds were created simultaneously in all wells in a confluent cell monolayer using 96-well Woundmaker Tool (Essen BioScience). After wounding, each well was carefully washed with culture media. Test wells for the invasion were covered with the ECM diluted to ~4 mg/mL and let to solidify in the incubator for 20 min. Tested compounds diluted to final concentration in 100 µL/well were added. Plate(s) were placed the IncuCyte®ZOOM system and images were taken with 10x objective with "Scratch Wound Wide Mode" scan type in phase contrast and red fluorescence channel every 2 h. Analysis was performed in IncuCyte® Scratch Wound Cell Migration Software Module (Cat. No. 9600−0012). Wound width (µm) or red cell count *per* wound is expressed as mean ± SD from six technical replicates.

**Co-culture of healthy donor neutrophils and chemoresistant EC cell line NLR-NT2 CisR.** Blood from healthy donors was collected by venous puncture using BD Vacutainer® blood collection set into 10 mL BD Vacutainer® Heparin

Tubes (367878, Becton Dickinson, UK) and used for neutrophil isolation or centrifuged at 1,600 x g for 10 min at 4°C to collect plasma. Neutrophils were isolated from whole blood by 1-Step Polymorphs (AN221725, Accurate Chemical & Scientific Corp, NY, USA) according to the protocol of the manufacturer with adaptations published elsewhere [56] and stained with CellbriteTM 488 for 15 min according to the instruction of the manufacturer (#30090, Biotium, CA, USA). Neutrophils were finally resuspended in 1 mL of phenol-red free Gibco™ RPMI 1640 Medium (11835030, Paisley, UK) supplemented with 10% of autologous plasma and counted on a DxFlex flow cytometer (Beckman Coulter, Brea, CA, USA).

Sextuplicates of NLR-NT2 CisR cells *per* each condition were plated at seeding density $1\times10^3$ cells/100 µL media per well in 96-well plates (TPP™ Cell Culture-Treated, Flat-Bottom Microplate) 24 h prior to coculture start. 50 µL of media was discarded from each well. Isolated neutrophils were diluted in RPMI medium supplemented with 20% autologous plasma to the concentration of $4\times10^6$ neutrophils/mL. 50 µL of suspension containing 200,000 neutrophils (E:T ratio 20:1) was added to designated wells, control wells were supplemented with medium only. After 1 h coincubation, wells were supplemented with treatments and plates were let to equilibrate for 30 min before the first scan in the IncuCyte®ZOOM system. Images were taken with 10x objective in phase contrast to monitor confluence and red fluorescence channel to monitor tumor cell proliferation every 2–3 h. Data were analyzed by IncuCyte®ZOOM software version 2016A. Data are expressed as mean±SD of the Total Red Object Integrated Intensity (RCU x µm²/Image).

**Therapeutic xenograft model.** The project was conducted at the Animal facility for immunodeficient mice of the Biomedical Research Center of the Slovak Academy of Sciences (BMC SAS), operating under license No. SK UCH 02017. The project was approved by the Institutional Ethics Committee of the BMC SAS and the State Veterinary and Food Administration of the Slovak Republic, the latter of which serves as the national competence authority. The project was registered under Nos. Ro1030/18–221 and 5862–3/2023–220 and conducted in accordance with the Directive 2010/63/EU and the Regulation 377/2012, which aim to ensure the protection of animals used for scientific purposes.

Six- to eight-week-old NSG mice (The Jackson Laboratory, Bar Harbor, ME, USA) were used for xenograft engraftment (n = 16 animals per one study, 2 studies in total).

In the first study of the recombinant human DNase (rhDNase) effect *in vivo*, suspension of $2\times10^6$ NT2 CisR cells was diluted in 100 µL of ECM:medium mixture 1:1 (50 µL serum-free DMEM plus 50 µL ECM) and injected into the flank s.c., with two injections *per* animal, in total n = 16 animals, 14 days prior to treatment start. Mice were randomly assigned to four study groups: 1. CPT i.p.; 2. rhDNase i.p.; 3. rhDNase i.p. and CPT i.p.; 4. untreated controls. Respective animals were treated with 4 doses of CPT (3 mg/kg/dose) once a week and/or with rhDNase Pulmozyme® 100 U/mouse for consequent 5 days with 2-day break. Xenografts were measured by caliper twice a week and their volume was calculated according to the formula for the volume of ellipsoid: $V = 0.52 \times ((width+length)/2)^3$. All animals were sacrificed when the xenografts in the control group exceeded volume 1 cm³. Xenografts at autopsy were excised, weighted, and fixed in 4% buffered formalin for subsequent immunohistochemistry (IHC) analysis. The tumor growth was evaluated as the mean of tumor volume or endpoint tumor weight.

In the second study focuses on overall survival, NT2 CisR cells were injected 10 days prior to treatment start (n = 16 animals/study) and treated and observed as described above. Animal was sacrificed at experiment endpoint when xenograft burden exceeded volume 1 cm³. Xenografts at autopsy were excised, macroscopically examined, and fixed in 4% buffered formalin for subsequent IHC analysis.

**IHC of xenografts from therapeutic experiment.** The slides were incubated in TRIS-EDTA retrieval solution (10 mM TRIS, 1 mM EDTA pH 9.0) at 97 °C for 20 min in the automated water bath heating process in Dako PT Link (Dako, Glostrup, Denmark) for tissue epitopes demasking. Slides were pre-treated with hydrogen peroxide for 5 min to block endogenous peroxidase activity. The slides were subsequently incubated for 20 min at RT with the primary monoclonal mouse antibodies against CD31 and CD34 (Dako, CD31: JC70A and CD34: QBEnd 10), Ready-to-Use and

immunostained using anti-mouse/anti-rabbit immuno-peroxidase polymer (EnVision FLEX/ HRP, Dako) for 20 min at RT, according to the instructions of the manufacturer. Color reaction was developed with diaminobenzidine substrate-chromogen solution (DAB, Dako, Glostrup, Denmark) for 10 min. Finally, the slides were counterstained with haematoxylin for 5 min.

**Intratumoral microvascular density.** Intratumoral microvascular density (iMVD) was determined as described previously [57]. Briefly, the "hot spot" area with high microvascular density under a low-power field of 40 times were identified, and then the number of CD31 and CD34 staining microvessels in this area was counted under a high-power field (HPF) of 400 times. Five HPF in the "hot spot" area were randomly selected, and the average value was taken after counting, which was the iMVD of the section [58,59].

**Statistical analyses.** The characteristics of patients were summarized using the mean or median (range) for continuous variables and frequency (percentage) for categorical variables, respectively. Statistical analysis was performed using non-parametric tests as the distribution of the ecDNA concentrations was significantly different from the normal distribution (Shapiro-Wilk test). The Mann-Whitney $U$ test was used for the analysis of the association of ecDNA to clinicopathological variables between the two groups of patients and Kruskal-Wallis test among more than two groups. Wilcoxon test was used to compare ecDNA before the 1st and 2nd cycle of chemotherapy.

Median follow-up period was calculated as a median observation time among all patients and among those still alive at the time of their last follow-up. Progression-free survival (PFS) was calculated from the date of the starting treatment with chemotherapy to the date of progression or death or the date of the last adequate follow-up. Over-all survival (OS) was calculated from the date of starting treatment with chemotherapy to the date of death or last follow-up. Survival rates were estimated using the Kaplan-Meier product limit method and were compared with the log-rank test to determine significance. Data about ecDNA concentrations in plasma including pellet microparticles and NETs-associated markers were dichotomized into high and low groups based on the ecDNA mean value of all samples. Plasma DNase level was dichotomized by the same approach using mean value of all samples. Multivariate Cox proportional hazards model for PFS and OS was used to assess differences in outcome based on ecDNA, DNase and prognosis according to IGCCCG (International Germ Cell Collaborative Group) in metastatic GCTs. All statistical tests were two-sided and significant $p$ values were set less or equal to 0.05. Statistical analyses were performed using NCSS 2007 software [60].

Analysis of data from animal experiments was conducted using SPSS software version 23 (IBM SPSS, Inc., Chicago, IL, USA). For *in vivo* experiments, 8 animals *per* treatment group were used. A two-way repeated measures ANOVA with Greenhouse-Geisser correction was employed to evaluate the effects of four treatment types over individual time points. For comparisons of tumor weight measurements, multivariate analysis one-way ANOVA test was used. Multiple comparisons were performed by Bonferroni test. Survival analysis was performed using the Kaplan-Meier method, with differences between survival curves analyzed by the log-rank test. $p$ values <0.05 were considered statistically significant. GraphPad Prism® software (GraphPad Inc.) was used to create the graphs.

## Sex as a biological variable

Our study exclusively examined male mice because the disease modeled is only relevant in males.

## Results

### Characteristics of the patients

The study population consisted of the 117 GCT patients and the 19 age-matched healthy donors (HDs). Median age of the patients was 34 years (range: 19–58 years) and of the healthy donors was 35 years (range: 26–50 years), ($p$ = 1.00). Patient characteristics are summarized in the Table 1.

**Table 1. Characteristics of patients.**

| Variable | N | % |
|---|---|---|
| **All** | 117 | 100.0 |
| **Histology** | | |
| Seminoma | 32 | 27.4 |
| Non-seminoma | 85 | 72.6 |
| **Disease primary** | | |
| Gonadal | 113 | 96.6 |
| Extragonadal | 4 | 3.4 |
| **Response to therapy** | | |
| Favorable | 99 | 84.6 |
| Unfavorable | 18 | 15.4 |
| **Disease stage** | | |
| I.A | 2 | 1.7 |
| I.B | 11 | 9.4 |
| I.S | 5 | 4.3 |
| II.A | 11 | 9.4 |
| II.B | 21 | 17.9 |
| II.C | 10 | 8.5 |
| III.A | 9 | 7.7 |
| III.B | 14 | 12.0 |
| III.C | 21 | 17.9 |
| **IGGCCG risk group** | | |
| Good | 61 | 52.1 |
| Intermediate | 10 | 8.5 |
| Poor | 20 | 17.1 |
| Stage I disease | 13 | 11.1 |
| Relapse | 13 | 11.1 |
| **Number of metastatic sites** | | |
| 0 | 18 | 15.4 |
| 1-2 | 70 | 59.8 |
| > 3 | 29 | 24.8 |
| **Localization of metastases** | | |
| Retroperitoneum | 96 | 82.1 |
| Mediastinum | 17 | 14.5 |
| Other lymphadenopathy | 19 | 16.2 |
| Lungs | 31 | 26.5 |
| Liver | 17 | 14.5 |
| Brain | 5 | 4.3 |
| Other metastases | 8 | 6.8 |
| **S stage** | | |
| 0 | 35 | 29.9 |
| 1 | 41 | 35.0 |
| 2 | 20 | 17.1 |
| 3 | 21 | 17.9 |

**Abbreviations:** IGCCCG—International Germ Cell Collaborative Group.

## Comparison of the ecDNA in GCTs patients to HDs

Detailed analysis of the plasma samples was focused on following analyses: DNase activity (K.U./mL), total extracellular ecDNA (ng/mL), nuclear ncDNA (GE/mL), mitochodrial mtDNA (GE/mL), pellet total ecDNA (ng/mL), pellet ncDNA (GE/mL), pellet mtDNA (GE/mL), all microparticles, subcellular particles according to their size < 100 nm, 100−500 nm, 500−1,000 nm, particles < 5 μm, particles > 5 μm, small particles (< 1 μm), large particles (> 1 μm); and NETs-associated markers myeloperoxidase MPO (ng/mL) and neutrophil elastase NE (ng/mL).

GCTs patients have significantly higher concentrations of the plasma ecDNA compared to the HDs (Table 2, S1 Table, S3 Fig). The mean concentration of plasma ecDNA in the HDs was 1.2 ng/mL (± 2.2), in the stage I patients 2.4 ng/mL (± 1.7), in the stage IS-III patients 6.6 ng/mL (± 0.7); and in the relapsed patients 7.5 ng/mL (± 1.8). Importantly, the enzymatic activity of the extracellular DNase in the plasma of the GCT patients was significantly decreased. The mean of DNase activity in the HDs was 1.5 K.U./mL, in the stage I patients 1.2 K.U./mL, in the stage IS-III patients 1.0 K.U./mL; and in the relapsed patients 0.8 K.U./mL. In addition, the NETs-associated markers exhibit increasing values with the disease stage. The mean MPO concentration in the HDs was 5.0 ng/mL, in the stage I patients 9.6 ng/mL, in the stage IS-III patients 9.9 ng/mL; and in the relapsed patients 12 ng/mL. The mean NE concentration in the HDs was 1.6 ng/mL, in the stage I patients 2.6 ng/mL, in the stage IS-III patients 3.2 ng/mL; and in the relapsed patients 2.1 ng/mL.

There were no differences in the ncDNA concentration in plasma nor in the amount of plasma microparticles between the GCTs patients and the HDs except for the ecDNA and mtDNA concentrations that were significantly higher in plasma from GCTs patients ($p = 0.00001$). Comparison between the stage I patients and the HDs showed no differences in the measured parameters, except for the plasma pellet mtDNA that was higher in the stage I patients compared to the HDs (median 63,162 GE/mL *vs.* 52,058 GE/mL, $p = 0.007$). There were no differences in the ecDNA concentration and NETs-associated markers between relapsed and newly diagnosed metastatic GCTs patients except for the plasma pellet mtDNA that was significantly higher in the relapsed compared to the newly diagnosed metastatic GCTs (median 679,379 GE/mL *vs.* 79,618 GE/mL, $p = 0.02$), whilst the DNase activity was lower in the relapsed GCTs (median 0.8 K.U./mL *vs.* 1.0 K.U./mL, $p = 0.02$).

## Association between the ecDNA and patients/tumor characteristics

We identified that the total plasma ecDNA concentration was the lowest in a good prognosis and the highest in a poor prognostic group ($p = 0.01$). No association between IGCCCG risk group and other parameters was found (S2 Table). The total plasma ecDNA and ncDNA increased with the number of metastatic sites, while the DNase activity decreased (S4 Fig). The amount of the total plasma DNA-containing pellet particles 100–500 nm, particles > 5 μm, small particles (< 1 μm), and large particles (> 1 μm) was associated with the number of metastatic sites (all $p < 0.05$) (S3 Table). Moreover, concentrations of the total ecDNA and ncDNA in plasma was positively associated with the treatment response – lower concentration was identified in the patients with favorable response to therapy, e.g., complete remission or partial remission with negative serum tumor marker (median total ecDNA 3.6 ng/mL *vs.* 10.0 ng/mL, $p = 0.007$; median ncDNA 2,884 GE/mL *vs.* 6,055 GE/mL, $p = 0.009$), while plasma DNase activity was higher in the GCTs patients that achieved favorable response (median 1.0 K.U./mL *vs.* 0.8 K.U./mL, $p = 0.03$) (S4 Table).

There were no differences in the DNase activity and ecDNA concentrations between the 1st and the 2nd cycle of therapy, while patients before the 2nd cycle of therapy had significantly higher concentration of pellet particles associated with DNA (< 100 nm, 100–500 nm and > 500 nm) as well as MPO (all $p < 0.05$).

## Prognostic value of ecDNA and DNase in GCTs patients

Next, we analyzed prognostic value of the plasma biomarkers associated with DNA and DNase for the GCT patients. In median follow-up of 22.6 months (range: 0.1–100.4 months), 27 (23.1%) patients experienced disease progression, and

**Table 2. Comparison of ecDNA and DNase in GCTs patients and HDs.**

| | N | Mean | Median | SD | SEM | *p* value |
|---|---|---|---|---|---|---|
| **Plasma total ecDNA ng/mL** | | | | | | |
| Stage I | 13 | 2.4 | 1.9 | 1.6 | 1.7 | **0.00001** |
| Stage IS – III | 70 | 6.6 | 4.1 | 6.9 | 0.7 | |
| Relapsed | 12 | 7.5 | 5.0 | 6.5 | 1.8 | |
| HDs | 8 | 1.2 | 1.1 | 0.9 | 2.2 | |
| **Plasma ncDNA ge/mL** | | | | | | |
| Stage I | 11 | 3388.3 | 3382.0 | 2056.6 | 2862.4 | 0.10257 |
| Stage IS – III | 64 | 7151.5 | 2924.5 | 11186.4 | 1186.7 | |
| Relapsed | 13 | 7170.2 | 5495.0 | 8810.8 | 2633.0 | |
| HDs | 15 | 2515.1 | 2232.0 | 2156.3 | 2451.2 | |
| **Plasma mtDNA ge/mL** | | | | | | |
| Stage I | 13 | 223274.2 | 77605.0 | 331687.9 | 51288.9 | 0.43953 |
| Stage IS – III | 71 | 164092.8 | 106748.0 | 171122.0 | 21946.5 | |
| Relapsed | 13 | 146052.5 | 91017.0 | 185318.5 | 51288.9 | |
| HDs | 19 | 85315.9 | 76536.0 | 51622.3 | 42424.6 | |
| **Plasma DNase k.u/mL** | | | | | | |
| Stage I | 13 | 1.2 | 1.1 | 0.5 | 0.1 | **0.00026** |
| Stage IS – III | 91 | 1.0 | 1.0 | 0.4 | 0.0 | |
| Relapsed | 13 | 0.8 | 0.8 | 0.3 | 0.1 | |
| HDs | 19 | 1.5 | 1.5 | 0.5 | 0.1 | |
| **Pellet total ecDNA ng/mL** | | | | | | |
| Stage I | 13 | 1.7 | 1.9 | 0.5 | 0.9 | 0.15069 |
| Stage IS – III | 67 | 3.0 | 1.9 | 3.8 | 0.4 | |
| Relapsed | 11 | 2.3 | 1.5 | 1.9 | 1.0 | |
| HDs | 19 | 1.9 | 1.3 | 2.2 | 0.7 | |
| **Pellet ncDNA GE/mL dich** | | | | | | |
| Stage I | 8 | 98613.6 | 14877.5 | 207901.8 | 91430.7 | 0.31692 |
| Stage IS – III | 56 | 92574.9 | 3997.5 | 310584.6 | 34557.6 | |
| Relapsed | 11 | 8656.5 | 2986.0 | 11092.4 | 77972.4 | |
| HDs | 15 | 43675.0 | 4164.0 | 100762.0 | 66771.6 | |
| **Pellet mtDNA GE/mL** | | | | | | |
| Stage I | 13 | 66347.8 | 26873.0 | 63161.9 | 237992.3 | **0.00001** |
| Stage IS – III | 72 | 336571.6 | 79618.0 | 765391.5 | 101127.3 | |
| Relapsed | 13 | 1174524.0 | 679379.0 | 1843476.0 | 237992.3 | |
| HDs | 18 | 27943.2 | 9906.5 | 52057.6 | 202254.6 | |
| **< 100 nm** | | | | | | |
| Stage I | 13 | 108876.9 | 43600.0 | 125685.1 | 42285.2 | 0.3678 |
| Stage IS – III | 72 | 102327.8 | 52500.0 | 114421.6 | 17967.8 | |
| Relapsed | 13 | 138369.2 | 28000.0 | 320473.3 | 42285.2 | |
| HDs | 8 | 52450.0 | 31900.0 | 52629.6 | 53903.3 | |
| **100-500 nm** | | | | | | |
| Stage I | 13 | 541600.0 | 395800.0 | 299063.5 | 241234.2 | 0.45587 |
| Stage IS – III | 72 | 713036.1 | 453900.0 | 840152.4 | 102504.8 | |
| Relapsed | 13 | 756784.6 | 239800.0 | 1437781.0 | 241234.2 | |
| HDs | 8 | 617950.0 | 503500.0 | 408755.3 | 307514.5 | |
| **500-1000 nm** | | | | | | |

*(Continued)*

**Table 2.** (Continued)

| | N | Mean | Median | SD | SEM | *p* value |
|---|---|---|---|---|---|---|
| Stage I | 13 | 868153.9 | 819800.0 | 483480.6 | 165734.2 | 0.46443 |
| Stage IS – III | 72 | 964094.4 | 854100.0 | 677975.8 | 70423.5 | |
| Relapsed | 13 | 997369.3 | 998000.0 | 248206.5 | 165734.2 | |
| HDs | 8 | 1016150.0 | 985000.0 | 186229.5 | 211270.4 | |
| **< 5 µM** | | | | | | |
| Stage I | 13 | 469138.5 | 266600.0 | 502673.7 | 369108.1 | 0.06704 |
| Stage IS – III | 72 | 1065272.0 | 245100.0 | 1534142.0 | 156840.8 | |
| Relapsed | 13 | 1213785.0 | 1263400.0 | 934771.0 | 369108.1 | |
| HDs | 8 | 194750.0 | 199900.0 | 68119.5 | 470522.3 | |
| **> 5 µM** | | | | | | |
| Stage I | 13 | 202553.8 | 86000.0 | 423417.4 | 186943.1 | 0.1063 |
| Stage IS – III | 72 | 420850.0 | 132100.0 | 633481.4 | 79435.5 | |
| Relapsed | 13 | 692876.9 | 214600.0 | 1140296.0 | 186943.1 | |
| HDs | 8 | 156625.0 | 126300.0 | 115749.2 | 238306.6 | |
| **Small particles (< 1 µM)** | | | | | | |
| Stage I | 13 | 1518631.0 | 1284200.0 | 756114.2 | 375895.1 | 0.90092 |
| Stage IS – III | 72 | 1779458.0 | 1470800.0 | 1389892.0 | 159724.7 | |
| Relapsed | 13 | 1892523.0 | 1457400.0 | 1851412.0 | 375895.1 | |
| HDs | 8 | 1686550.0 | 1540500.0 | 561741.4 | 479174.1 | |
| **Large particles (> 1 µM)** | | | | | | |
| Stage I | 13 | 671692.3 | 410400.0 | 802725.8 | 516065.8 | 0.09862 |
| Stage IS – III | 72 | 1486122.0 | 370800.0 | 2050059.0 | 219285.8 | |
| Relapsed | 13 | 1906662.0 | 1463800.0 | 1974916.0 | 516065.8 | |
| HDs | 8 | 351375.0 | 319800.0 | 175239.3 | 657857.4 | |
| **All particles** | | | | | | |
| Stage I | 13 | 2190323.0 | 2109200.0 | 1059819.0 | 699659.1 | 0.50064 |
| Stage IS – III | 72 | 3265581.0 | 2093300.0 | 2780498.0 | 297298.0 | |
| Relapsed | 13 | 3799185.0 | 3092600.0 | 2654344.0 | 699659.1 | |
| HDs | 8 | 2037925.0 | 1912400.0 | 556671.3 | 891893.9 | |
| **MPO (ng/mL)** | | | | | | |
| Stage I | 13 | 9.6 | 5.8 | 10.2 | 2.3 | **0.01354** |
| Stage IS – III | 73 | 9.9 | 6.9 | 8.1 | 1.0 | |
| Relapsed | 13 | 12.9 | 7.9 | 12.7 | 2.3 | |
| HDs | 19 | 5.0 | 4.4 | 2.5 | 1.9 | |
| **NE (ng/mL)** | | | | | | |
| Stage I | 10 | 2.6 | 2.6 | 1.3 | 0.7 | **0.01117** |
| Stage IS – III | 57 | 3.2 | 2.4 | 2.4 | 0.3 | |
| Relapsed | 9 | 2.1 | 1.7 | 1.7 | 0.7 | |
| HDs | 13 | 1.6 | 1.2 | 1.2 | 0.6 | |

**Abbreviations:** ecDNA, extracellular DNA, ncDNA, nuclear DNA, mtDNA, mitochondrial DNA, MPO, myeloperoxidase, NE, neutrophil elastase, SD, standard deviation, SEM, standard error of mean.

21 (18.0%) patients died. Patients with lower baseline level of plasma total ecDNA, ncDNA and pellet mtDNA had significantly better PFS, and patients with lower plasma ecDNA, ncDNA, pellet ecDNA and mtDNA had significantly better OS. Inversely, higher plasma DNase level was associated with better PFS and OS (Table 3, Fig 1A-1D). There were no differences ecDNA, DNase activity and NETs-associated markers before the 2nd cycle of chemotherapy and the outcome of patients. In multivariate analysis, prognostic value of ecDNA and DNase was not independent of IGCCCG risk category.

## Association between ecDNA, DNA damage in PBMC and inflammatory indexes

Next, we analyzed if the DNase level in plasma was associated with other plasma parameters, such as a level of the DNA damage determined by the comet assay in PBMCs, or systemic inflammation. Patients were dichotomized according to the mean DNase activity to 'DNase low' and 'DNase high' group. Patients with lower DNase activity exhibited significantly higher DNA damage level, NLR and SII inflammatory indexes. Patients with lower DNase activity exhibited significantly lower MLR values (Table 4).

## Association between ecDNA and specific immune cell subpopulations

Analysis of possible associations between ecDNA and different innate immune cells percentage in GCT patients showed that plasma total ecDNA and ncDNA positively correlated with neutrophils, classical monocytes and polymorphonuclear monocytes (PNMs) percentage, while the inverse correlation was observed between plasma total ecDNA and ncDNA and nonclassical monocytes and basophils percentage, respectively. Moreover, total ecDNA in plasma higher than mean was associated with decreased conventional dendritic cells (cDCs) and plasmocytoid dendritic cells (pDCs) percentage. The inverse association was also determined between plasma total ecDNA and ncDNA and percentage of CD1c positive cells within DCs (all $p < 0.05$).

**Table 3. Association between ecDNA and outcome of patients.**

| Variable | PFS | | | | OS | | | |
|---|---|---|---|---|---|---|---|---|
| | HR | Lower 95% CI | Upper 95% CI | *p* value | HR | Lower 95% CI | Upper 95% CI | *p* value |
| Plasma total ecDNA ng/mL | 0.35 | 0.12 | 1.00 | **0.015303** | 0.28 | 0.09 | 0.92 | **0.009299623** |
| Plasma ncDNA GE/mL | 0.31 | 0.1 | 0.93 | **0.005226** | 0.31 | 0.09 | 1.06 | **0.012502708** |
| Plasma mtDNA GE/mL | 0.7 | 0.26 | 1.87 | 0.426150 | 0.86 | 0.32 | 2.3 | 0.765290449 |
| Plasma DNaseUK.u./mL | 2.96 | 1.39 | 6.32 | **0.013022** | 4.88 | 2.06 | 11.53 | **0.004673036** |
| Pellet total ecDNA ng/mL | 0.28 | 0.09 | 0.93 | **0.005225** | 0.34 | 0.1 | 1.12 | **0.026515578** |
| Pellet ncDNA GE/mL mean | 0 | 0 | 0.00 | 0.158257 | 0 | 0 | 0 | 0.28296951 |
| Pellet mtDNA GE/mL | 0.13 | 0 | 5.35 | **0.000996** | 0.1 | 0 | 6.72 | **0.00009945** |
| Microparticles < 100 nm | 0.74 | 0.28 | 1.95 | 0.511062 | 0.49 | 0.16 | 1.47 | 0.147607302 |
| Microparticles 100–500 nm | 1.36 | 0.5 | 3.73 | 0.574703 | 1.06 | 0.35 | 3.24 | 0.913891172 |
| Microparticles 500–1000 nm | 1.11 | 0.45 | 2.75 | 0.822657 | 1.15 | 0.42 | 3.11 | 0.790837587 |
| Microparticles < 5 μM | 0.49 | 0.18 | 1.35 | 0.101592 | 0.6 | 0.21 | 1.72 | 0.291563971 |
| Microparticles > 5 μM | 0.5 | 0.2 | 1.20 | 0.111855 | 0.47 | 0.18 | 1.25 | 0.145727291 |
| Small particles (< 1 μM) | 1.01 | 0.42 | 2.43 | 0.989093 | 0.88 | 0.33 | 2.36 | 0.805370795 |
| Large particles (>1 μM) | 0.67 | 0.26 | 1.72 | 0.363855 | 0.83 | 0.31 | 2.27 | 0.709872806 |
| All particles | 0.72 | 0.27 | 1.90 | 0.472192 | 0.96 | 0.35 | 2.66 | 0.94047604 |
| Dichotomized myeloperoxidase (ng/mL) | 0.67 | 0.26 | 1.74 | 0.385108 | 0.77 | 0.27 | 2.21 | 0.6119754 |
| Neutrophil elastase (ng/mL) | 1.43 | 0.52 | 3.89 | 0.507157 | 2.01 | 0.65 | 6.17 | 0.27413669 |

**Abbreviations:** PFS, progression-free survival, OS, overall survival, HR, hazard ratio, CI, confidence interval, SD, standard deviation, SEM, standard error of mean.

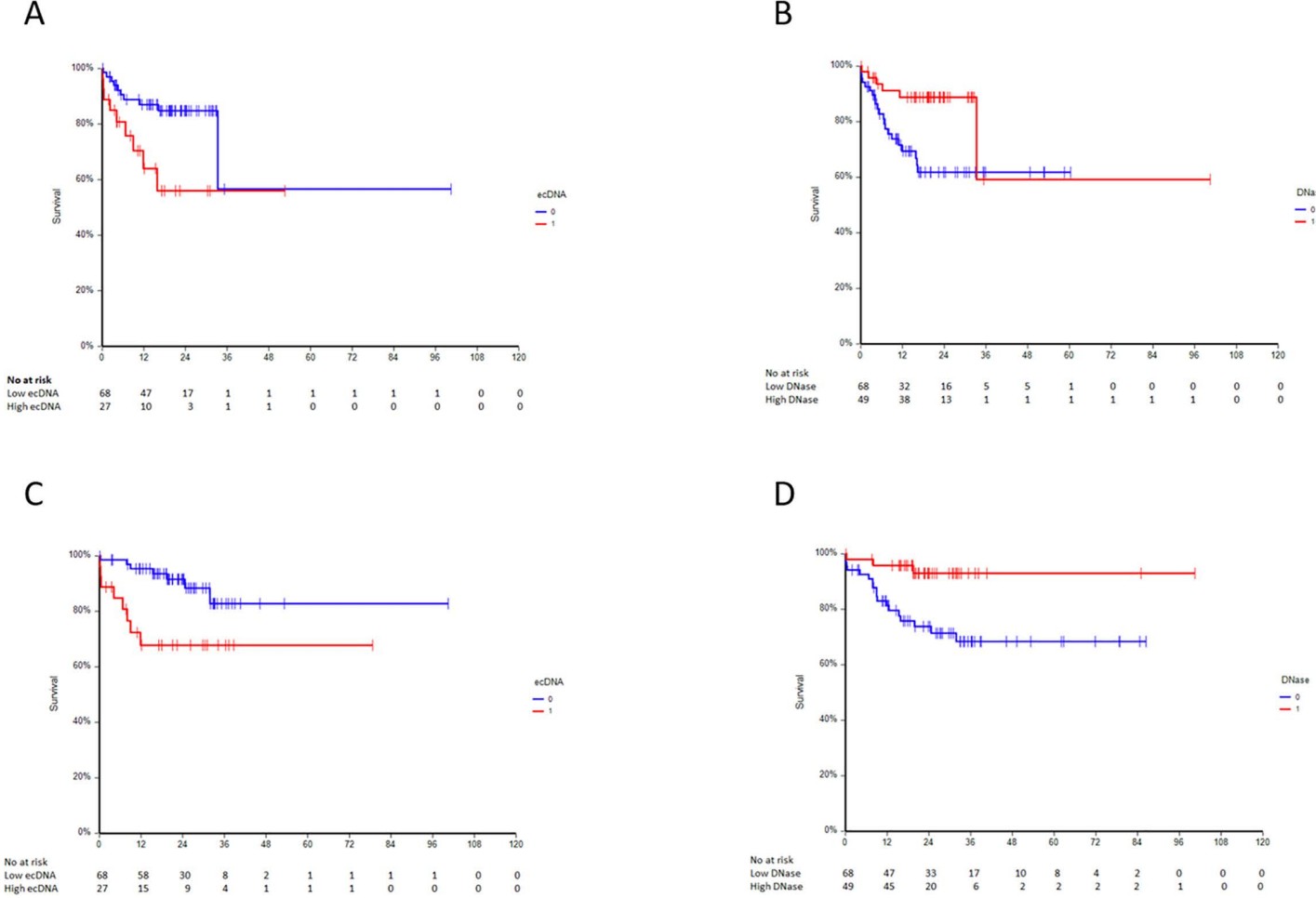

**Fig 1. (A) Kaplan–Meier estimates of probabilities of PFS in GCT patients according to plasma ecDNA, HR = 0.35, 95% CI 0.12-1.00, p = 0.02.** (B) Kaplan–Meier estimates of probabilities of PFS in GCT patients according to plasma DNase activity, HR = 2.96, 95% CI 1.39-6.32, p = 0.01. (C) Kaplan–Meier estimates of probabilities of OS in GCT patients according to plasma ecDNA, HR = 0.28, 95% CI 0.09-0.92, p = 0.01. (D) Kaplan–Meier estimates of probabilities of OS in GCT patients according to plasma DNase activity, HR = 4.88, 95% CI 2.06-11.53, p = 0.005.

Assessing of selected adaptive immune cells subpopulations showed significant link only between plasma total ecDNA and ncDNA and lymphocytes percentage, when plasma total ecDNA and ncDNA higher than mean correlated with lower lymphocytes percentage.

Furthermore, analyzed GCT patients with DNase activity above the cut-off value had significantly lower mean percentage of neutrophils ± standard error of the mean (SEM) 60.5 ± 1.9% when compared to patients with DNase activity under the cut-off, 65.8 ± 1.9% ($p = 0.028$). Similarly, the negative correlation was also showed between DNase activity and CD16 + HLADR+Lin- DCs percentage. On the other hand, DNase was positively associated with CD1c positive cells within DCs and lymphocytes percentage (all $p < 0.05$).

### Association between DNA-containing pellet particles in plasma and specific immune cell subpopulations

Concentration of DNA-containing plasma pellet particles < 5 µm higher than cut-off was significantly associated with decreased monocytes, eosinophils, basophils and NKT cells percentage. Contrarily, high level of DNA-containing plasma

**Table 4. Association between DNase activity and DNA damage in PBMCs and inflammatory indexes.**

| Variable | N | Mean | Median | SD | SEM | *p* value |
|---|---|---|---|---|---|---|
| **DNA damage** | | | | | | |
| DNAase low | 59 | 6.0 | 5.6 | 2.4 | 0.4 | **0.03279** |
| DNAase high | 43 | 5.6 | 4.5 | 3.6 | 0.5 | |
| **NLR** | | | | | | |
| DNAase low | 56 | 6.2 | 4.0 | 10.0 | 1.1 | **0.00712** |
| DNAase high | 30 | 3.8 | 2.3 | 3.8 | 1.5 | |
| **MLR** | | | | | | |
| DNAase low | 56 | 1.5 | 1.4 | 0.6 | 0.1 | **0.04926** |
| DNAase high | 30 | 1.7 | 1.7 | 0.6 | 0.1 | |
| **SII** | | | | | | |
| DNAase low | 56 | 1923.3 | 926.8 | 3774.9 | 436.8 | **0.0235** |
| DNAase high | 30 | 1359.0 | 569.1 | 1981.4 | 596.8 | |

**Abbreviations:** NLR, neutrophil to lymphocyte ratio, MLR, monocyte to lymphocyte ratio, SII, systemic inflammatory index, SD, standard deviation, SEM, standard error of mean.

pellet particles < 5 µm correlated with increased classical monocytes and NK cells percentage. Analysis of DNA-containing plasma pellet particles >5µm revealed only positive association with PNMs percentage.

Small plasma pellet particles associated with DNA < 1µm positively correlated with PNMs percentage. Furthermore, the negative association was described between this type of plasma pellet particles and B cells and $T_{reg}$ percentage, while analysis of pellet particles >1µm showed negative association with basophils and NKT cell percentage. Moreover, besides the positive correlation of pellet particles >1µm with classical monocytes percentage, also the similar link with PNMs percentage was showed.

Small DNA-containing plasma pellet particles < 100 nm significantly positively correlated with classical monocytes and PNMs percentage, while GCT patients with plasma pellet particles < 100 nm above the cut-off had lower NK cells and cDCs percentage. Assessing DNA-containing plasma pellet particles 100−500 nm revealed only positive correlation with PNMs. Concentration of DNA-containing plasma pellet particles 500−1,000 nm was positively associated with CD4 + NKT cells and CD8 + NKT cells percentage as well as with T cytotoxic cells percentage. On the other hand, patients with plasma pellet particles 500−1,000 nm above cut-off value had decreased B cells percentage compared with patients with plasma pellet particles 500−1,000 nm under the cut-off.

## Association between NETs associated markers and specific immune cell subpopulations

Analysis of the possible link between NETs and changes in innate immune cells subpopulations showed that NETs are inversely associated with monocytes percentages. Similar association was also found between NETs and NKT cells percentage. Contrarily, patients with NETs under the cut-off had lower classical monocytes and PNMs percentage. In addition, there was also the inverse correlation between MPO and nonclassical monocytes and PNMs percentage.

## The effect of DNase I on GCT cell line proliferation, chemoresistance and migration

Based on the growing body of evidence, the use of a DNase for improvement of efficacy of cytostatic and/or cytotoxic chemotherapeutic compounds in solid tumors was suggested [61,62]. Search for any information about the treatment with DNase I as a single agent or in combination in GCT model and/or patient did not reveal any experimental data. Therefore, we designed the preclinical study to evaluate the effect of DNase I treatment alone and in combination with CDDP on the xenografts derived from the human embryonal carcinoma (EC) chemoresistant cell line variant NT2 CisR.

In our preclinical study, we tested the effect of two different DNase I enzymes: either isolated from bovine pancreatic (bpDNase) or recombinant human enzyme rhDNase. $IC_{10}$ for bpDNase was achieved at ~500 K.U./mL in EC cell lines NCCIT, Tera-2 and NT2 CisR. Toxicity of bpDNase was higher in T/TC cells SuSa with $IC_{10} \geq 100$ K.U./mL (Fig 2A). Next, we decided to test effect of the rhDNase and its clinically approved formulation Pulmozyme® on the cells *in vitro*. $IC_{10}$ was not achieved with the concentrations up to 1,000 U/mL in the normal fibroblasts propagated from the human testis Hs1. Tes or chemoresistant EC cells NT2 CisR (Fig 2B). Treatment with rhDNase does not affect the response of NT2 CisR cells to CDDP (Fig 2C).

To further investigate the effect of rhDNase on CDDP-resistant cells, we employed live cell-imaging technology. We stably transduced model EC cells NT2 and NT2 CisR to express red nuclear fluorescence mKate2 and verified the absence of any effect on cell proliferation in the presence of the rhDNase. Lentiviral transduction also did not alter the chemoresistance of CDDP-resistant cells NLR-NT2 CisR. Live cell monitoring confirmed the absence of invasive capacity and minimal migratory capacity of the CDDP-resistant NLR-NT2 CisR cells: no significant effect of rhDNase was observed when comparing kinetics of the wound width closure in the standard scratch wound assay (Fig 2D). Recent reports suggested that neutrophils and the NET formation counteracted chemotherapy efficacy, which could be reverted by DNa-seI [39]. Thus, we were interested, if the rhDNase and CDDP combination treatment will be affected in the presence of human neutrophils. We performed direct coculture experiment of the NLR-NT2 CisR target tumor cells with neutrophils at effector:target ratio 20:1. After mimicking their direct interaction the treatments were added, and the proliferation of the tumor cells was monitored over the time as integrated total red fluorescence *per* well. There was no significant effect on the tumor cell proliferation in the presence of or neutrophils alone or 200 U/mL rhDNase as expected. On the other hand, significant inhibition was executed by cytotoxic drug CDDP at 0.5 μg/mL corresponding to $IC_{50}$ (CDDP) in the presence of neutrophils. The most important finding was the significant augmentation of cytotoxic effect of the cisplatin by 200 U/mL of rhDNase even in the cisplatin resistant EC cell line variant NLR-NT2 CisR (Fig 2E).

## The effect of rhDNase I on CDDP response *in vivo* on chemoresistant EC xenograft mouse model

In a preliminary experiment, it was proved that xenograft growth does not affect DNase activity (S5 Fig). In a subsequent therapeutic experiment, NSG mice (males, n = 16) bearing bilateral tumors were randomized to 4 groups: control, CDDP alone, Pulmozyme® alone or the combination of the two. Respective animals were treated 4 times with CDDP (3 mg/kg once a week) and/or with rhDNase I Pulmozyme® 100 U/mouse intraperitoneally (IP) for consequent 5 days with a 2-day break (Fig 3A). Animals were monitored daily and the tumor volumes were measured using a caliper twice a week. The experiment was terminated when the xenografts in the control group exceeded the volume of 1 cm³. All animals were sacrificed, tumors excised during autopsy and endpoint tumor weight was determined. Median tumor weight at the experiment endpoint in the DNase I treated animals was not significantly different from the untreated controls (206 mg *vs*. 308 mg, respectively). However, the combined treatment resulted in the median tumor weight of 49 mg, which was significantly lower in comparison to control ($p \geq 0.01$) (Fig 3B-3C). In the following survival experiment monitoring animal survival, we confirmed the effect of combined treatment on decreased tumor volume and significant prolongation of animal survival in the CDDP + Pulmozyme® treated group. The experiment was designed as depicted in Fig 3A. Tumor volumes were monitored twice a week by caliper measurements. The animal was sacrificed, when the xenograft volume exceeded 1 cm³. Growth of individual xenografts and evaluation of survival in Kaplan-Meier plot is depicted (Fig 3D-3E). Spaghetti plot depicts the delay in individual xenograft growth in the group treated with combination of CDDP + rhDNase and the significant prolongation of animal survival was confirmed (Log-rank test, $p = 0.035$). During the experiment trained personnel monitored animal behavior and no signs of toxicity were observed. At the time of sacrifice, each animal was autopsied and macroscopic evaluation of all internal organs did not show any abnormalities. We did not observe any toxicity related to this treatment combination.

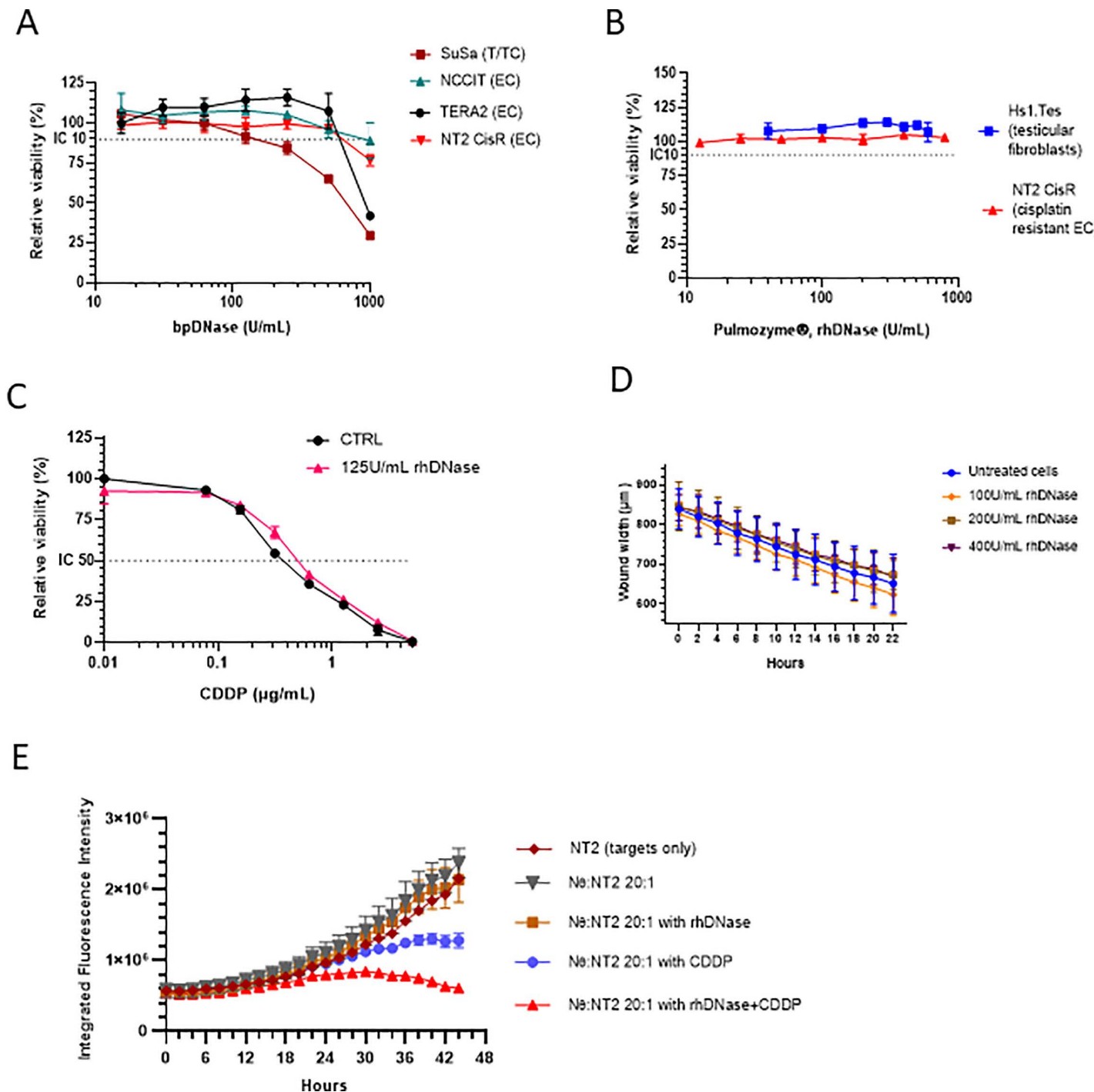

**Fig 2. Effect of DNase *in vitro*.** (A) Effect of bpDNase on GCT proliferation. (B) Pulmozyme® rhDNase does not inhibit proliferation of healthy human testicular fibroblasts or EC cells NT2. (C) DNase does not affect the chemoresistance of the EC cells NT2 CisR. (D) NLR-NT2 CisR cells retain low migratory potential in the presence of rhDNase *in vitro*. (E) Tumor cell chemosensitivity to cisplatin is significantly higher in the direct coculture of healthy donor Nθ in the presence of the rhDNase.

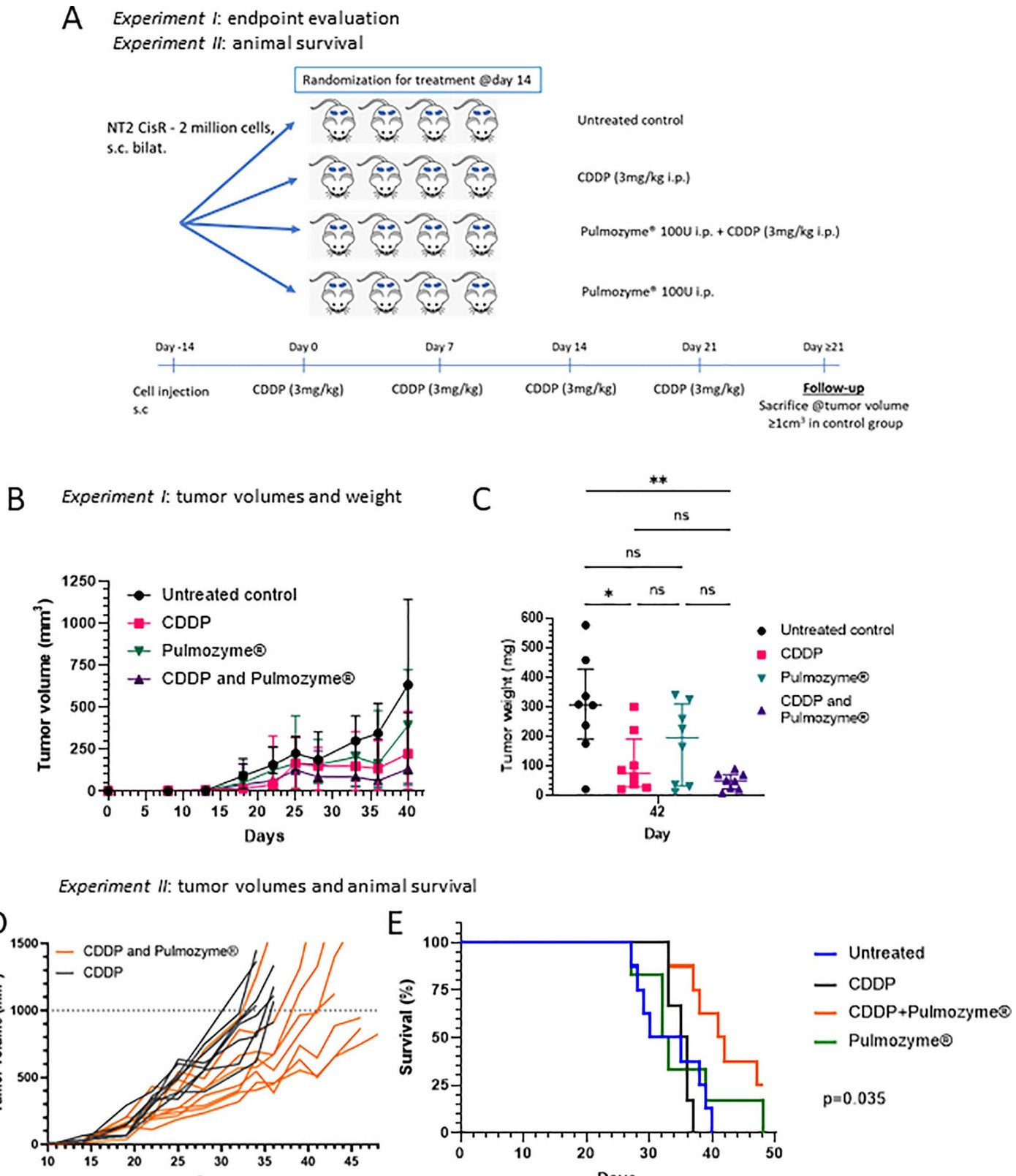

**Fig 3. Tumor growth control by rhDNase Pulmozyme® and CDDP combination treatment.** (A) Treatment scheme for two independent studies (n = 16 animals per study). (B) Tumor growth monitored using caliper measurements in the endpoint study. (C) Pulmozyme® rhDNase in combination

with CDDP administration significantly inhibits tumor growth in comparison to control as illustrated by the difference in the tumor wight at the experiment endpoint. (D) In the second – survival – study, the spaghetti plot depicting xenograft growth kinetics illustrates delayed tumor growth in the group treated with combination therapy (n = 4 mice pre group, n = 8 xenogtafts per group). (E) Survival plot illustrates significantly increased animal survival in the group (n = 4 mice per group) treated with the combination of Pulmozyme® and CDDP.

### Pulmozyme® treatment decreases intratumoral microvascular density in the xenograft model

The intratumoral microvascular density (iMVD), assessed by CD31 and CD34 immunohistochemical staining, was significantly lower in xenograft tumors treated with DNase compared to tumors without DNase exposure (Fig 4A–4B). For CD31, the mean ± SD iMVD was 27.41 ± 19.40 in DNase-exposed tumors (Pulmozyme® alone or in combination with CDDP) versus 49.60 ± 19.85 in non-exposed tumors (control and CDDP alone), p < 0.00001. For CD34, the mean ± SD iMVD was 40.73 ± 28.00 in DNase-exposed tumors compared with 67.96 ± 18.63 in non-exposed tumors, p < 0.00001 (Fig 4C).

## Discussion

A prognostic value of ecDNA concentrations was shown for various clinical settings in different types of cancer [63–69]. In GCTs, Boublikova et al. showed that total ecDNA were significantly higher in GCTs patients but without a clear threshold separating healthy and cancer samples [69]. Contrary, Ellinger et al. observed that ecDNA, mtDNA as well as hypermethylated ecDNA are higher in patients with testicular cancer with an accurate discrimination from healthy individuals [63–65]. A meta-analysis of studies on ecDNA in GCTs patients showed, that the sensitivity of ecDNA for GCTs was found to be higher than the sensitivity of serum tumor markers but lower than miR-371a-3p, with comparable specificity. Methylation analysis of ecDNA has shown clinical utility to accurately detect teratoma in GCTs patients [66]. In this translational study, we show higher plasma ecDNA levels in patients with GCTs compared to HDs. In addition, we demonstrate the prognostic value of the total ecDNA, ncDNA, but not mtDNA. Moreover, the total ecDNA correlated with the IGCCCG risk score, as well as with the metastatic burden. We report for the first-time the prognostic value of pre-treatment plasma ecDNA, including its subtypes as ncDNA, plasma microparticle-associated ecDNA and mtDNA. We revealed an inverse relationship between DNase activity and ecDNA, as well as between DNase activity, response to therapy and treatment outcome. Surprisingly, we did not observe differences in DNase activity and ecDNA concentrations between 1st and 2nd cycle of therapy and/or any prognostic value of ecDNA or DNase activity before the 2nd cycle of therapy. Unfortunately, methylation status of ecDNA was not assessed in our trial.

The association between systemic inflammatory markers, tumor characteristics and outcome of patients were evaluated in several studies including by our group, that consistently showed association between inflammatory markers and disease stage and prognosis [52,53,70,71]. Significantly different pattern of immune cell distribution was described in GCTs compared to normal testis [72]. Moreover, a unique cytokine profile characterized by higher concentrations of interleukin 6 and other B-cell supporting or T-helper cell type 1 driven cytokines was described in germ-cell neoplasia in situ and seminoma samples, but not in normal spermatogenesis samples [72]. In an exploratory analysis, we observed positive correlation between ecDNA including some of its subtypes and SIIs, while DNase activity exhibits an inverse correlation. We disclosed correlation between ecDNA, DNase activity and various immune cell populations. Beyond specific immune cell population, ecDNA correlated with lower blood lymphocytes percentage, while higher DNase activity correlated with lower percentage of neutrophils. We suggest that our data showing inverse association between ecDNA and DNase, their association with response to therapy and treatment outcome, in connection with their correlations to changes in immune cell populations and inflammatory indexes, support our hypothesis of connection between ecDNA, DNase, systemic inflammation and disease progression in GCTs.

Based on these observations, we hypothesize that DNase activity might modulate the biology of ecDNA in GCTs. We suppose that degradation of ecDNA by DNase might contribute to a better control of the tumor progression by CDDP in patients with progressing or relapsing GCTs. Previous studies using murine tumor models have proved the high

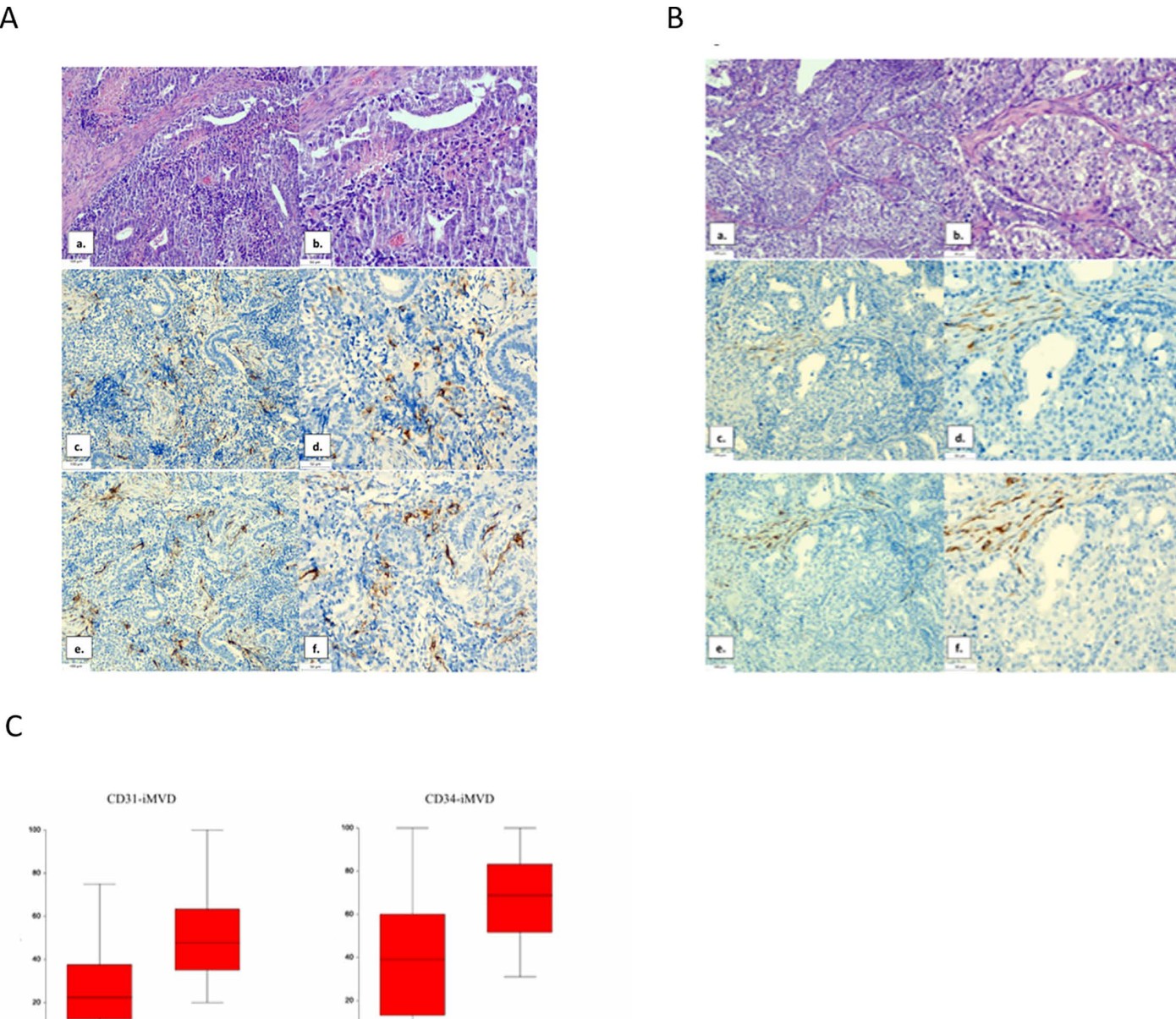

**Fig 4. Histological appearance of NT2 CisR xenograft tumors without (A) and with (B) DNase treatment, and quantification of intratumoral microvascular density (C).** (A) Tumors without DNase are composed of embryonal carcinoma with atypical cells showing enlarged nuclei and prominent nucleoli, together with epithelial glandular structures of teratoma (a, b, hematoxylin-eosin). Immunohistochemical staining for endothelial markers CD31 (c, d) and CD34 (e, f) shows abundant positive staining (brown) in endothelial cells of intratumoral blood vessels, indicating higher microvascular density. (B) Tumors with DNase show a similar histological composition, consisting of embryonal carcinoma with atypical cells and glandular teratoma structures (a, b, hematoxylin-eosin). In contrast to untreated tumors, immunohistochemical staining for CD31 (c, d) and CD34 (e, f) reveals a visibly lower number of positively stained blood vessels, reflecting reduced intratumoral microvascular density. (C) Box plot of intratumoral microvascular density based on CD31 and CD34 immunohistochemical staining confirms a decrease in microvascular density in DNase-treated tumors compared with untreated controls.

anti-metastatic potential of DNase I [43]. A study using Lewis lung carcinoma, melanoma B16, and lymphosarcoma RLS40 murine tumor models, revealed that tumor progression is accompanied by an increase in the level of SINEs and LINEs in the pool of circulating ecDNA. Treatment with DNase I decreased the number and area of metastases by factor 3–10, and the size of the primary tumor node by factor 1.5–2, which correlated with 5- to 10-fold decrease in SINEs and LINEs levels [43,73]. Another trial demonstrated that systemic treatment with DNase I and a protease mix in rats decreases ecDNA and proteins with subsequent antitumor effects [44].

Study by Mousset et al. suggested the utility of DNase I for improvement of efficacy of CDDP and adriamycin/cyclo-phosphamide in experimental model of breast cancer metastasis [28,39]. More importantly, rhDNase is available for clinical use (Pulmozyme®, Roche – Dornase alfa recombinant human deoxyribonuclease I, formulated 1 mg/1000 K.U./mL). Search for any information about the treatment with DNase I as a single agent or as part of combined modality in GCT model and/or patient did not reveal any experimental data. Therefore, we examined the effect of DNase I in vitro on selected GCT cell line models and subsequently we designed the preclinical proof-of-concept study to evaluate the effect of DNase I treatment alone and in combination with CDDP on the xenografts derived from the chemoresistant variant of the human EC cell line NTERA-2 CisR.

We used a previously established xenograft model setup employing immunocompromised NSG mice as the host for the NTERA2 CisR xenografts because they retain their innate neutrophil function and are often used for this type of pre-clinical studies [55,74,75]. In this immunodeficient NSG mouse model due to the high sensitivity to cytotoxic agents, the maximum tolerated dose of CDDP of 3 mg/kg once a week was used as previously published [76]. DNase I was used in multiple studies previously with the most common route of i.p. administration [38–40,77,78]. Administration via i.p. injections of drugs in experimental studies involving rodents is a justifiable route for pharmacological and proof-of-concept studies where the goal is to evaluate the effects of target engagement [79]. Based on the previously cited literature where the daily i.p. injections of DNase I range from 50–375 U/mouse, we decided to use the dose of 100 U/mouse of Pulmo-zyme® as the highest treatment dose used in the study by Alekseeva et al. [28].

Our experiments using selected model human GCT cell lines confirmed the lack of cytotoxic or inhibitory effects of DNase I in vitro. In vivo experiment showed that the combined treatment with DNase I and CDDP substantially delayed the growth of the NTERA-2 CDDP-resistant xenografts and reduced microvessel density in the xenografts. It has been described that CDDP treatment at maximum-tolerated dose increased microvessel density in the mouse model of human bladder cancer [80]. Moreover, chemotherapy treatment alone induces the formation of NETs [81,82]. Of note, NETs accelerated growth of gastric cancer cells by increasing and promoting angiogenesis, that could be abolished by DNase I treatment [77]. In our model, DNase I treatment reduced significantly microvessel density thereby counteracting the CDDP-induced NETs formation and CDDP-promoted tumor angiogenesis.

## Conclusions

In conclusion, the prominent effect observed in the preclinical study in vivo warranties the use of DNase I for combination therapy in GCTs and potentially other solid tumors for the better control of tumor growth. Based on the preclinical data, we suggest that there is a strong rationale to add DNase I to CDDP-based chemotherapeutic regimen in refractory GCTs. We hypothesize that supplementation of DNase I in patients will enable degradation of ecDNA and NETs, thus establishing CDDP sensitivity in patients with progressing or relapsing GCTs [27]. Neutrophils have emerged as pivotal contributors to the progression of cancer, with their role intricately linked to the formation of NETs, scaffolds of DNA associated with enzymes and proteases that are released in the extracellular space. Neutrophils per se and NETs shape response to cancer treatment across chemotherapy, radiotherapy, immunotherapy, and targeted therapy. Targeting neutrophils and NETs presents a strategic opportunity to design combination therapies, offering potential solutions to either overcome treatment resistance or to enhance the efficacy of anticancer interventions.

## Supporting information

**S1 Fig. Gating strategy.** All events were gated according to time for one minute which corresponds to 10 μL of sample volume. Calibration beads and blank sample were then used to create specific size gates and set the threshold on FITC-H channel for noise. Particles divided into small and large events were then separated according to size, as shown above.
(TIF)

**S2 Fig. (A) DNase vs. days of sample storage: Spearmen correlation = 0.3023, p = 0.0009.** (B) ecDNA vs. days of samples storage: Spearmen correlation = 0.0874, p = 0.3996.
(TIF)

**S3 Fig. Comparison of ecDNA and DNase in GCTs patients according to stage (A, B) and number of metastatic sites (C, D).**
(TIF)

**S4 Fig. The inverse correlation (Spearman correlation = −0.2021, p = 0.0495) between DNase activity and extracellular DNA.**
(TIF)

**S5 Fig. A preliminary experiment showing no changes in endogenous deoxyribonuclease activity in mice bearing growing xenografts.**
(TIF)

**S1 Table. Comparison of cfDNA ecDNA, DNase and markers of NETosis levels in GCTs patients and HDs.**
(DOCX)

**S2 Table. Association between ecDNA, DNase, markers of NETosis and IGGGCCG risk group.**
(DOCX)

**S3 Table. Association between ecDNA, DNase, markers of NETosis and number of metastatic sites.**
(DOCX)

**S4 Table. Association between ecDNA, DNase, markers of NETosis and treatment response.**
(DOCX)

**S5 Table. Association between ecDNA, DNase, markers of NETosis and specific immune cell subpopulations.**
(DOCX)

## Acknowledgments

We would like to acknowledge Mrs. Slackova, from the Population Registry of Slovak Republic for help with up-dating the patient database and Dr. Maria Reckova for discussions and critical input. We would like to acknowledge Dr. Daniela Svetlovska for administration support. We appreciate excellent laboratory assistance of Emilia Klincova, Mgr. Lucia Donarova and Bc. Lucia Rojikova.

## Author contributions

**Conceptualization:** Michal Mego, Lucia Kucerova, Peter Celec.

**Data curation:** Barbora Vlkova, Katarina Kalavska.

**Formal analysis:** Michal Mego, Lucia Kucerova, Peter Celec.

**Funding acquisition:** Michal Mego, Lucia Kucerova, Peter Celec.

**Investigation:** Michal Mego, Barbora Vlkova, Katarina Kalavska, Michal Pastorek, Zuzana Cierna, Zuzana Sestakova, Miroslav Chovanec, Natalia Udvorkova, Lucia Kucerova, Peter Celec.

**Methodology:** Barbora Vlkova, Katarina Kalavska, Michal Pastorek, Zuzana Cierna, Zuzana Sestakova.

**Project administration:** Michal Mego.

**Resources:** Michal Mego, Miroslav Chovanec, Lucia Kucerova, Peter Celec.

**Validation:** Michal Mego, Lucia Kucerova, Peter Celec.

**Visualization:** Michal Mego, Barbora Vlkova, Michal Pastorek, Lucia Kucerova, Peter Celec.

**Writing – original draft:** Michal Mego, Barbora Vlkova, Lucia Kucerova, Peter Celec.

**Writing – review & editing:** Michal Mego, Barbora Vlkova, Katarina Kalavska, Michal Pastorek, Zuzana Cierna, Zuzana Sestakova, Miroslav Chovanec, Natalia Udvorkova, Lucia Kucerova, Peter Celec.

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
