## [Decision Letter · Decision Letter 0]

5 Sep 2025

Dear Dr. Michal,

Thank you for submitting your manuscript to PLOS ONE. After careful consideration, we feel that it has merit but does not fully meet PLOS ONE’s publication criteria as it currently stands. Therefore, we invite you to submit a revised version of the manuscript that addresses the points raised during the review process.

We look forward to receiving your revised manuscript.

Kind regards,

Milad Khorasani, PhD

Academic Editor

PLOS ONE

**Journal Requirements:**

1. When submitting your revision, we need you to address these additional requirements. Please ensure that your manuscript meets PLOS ONE's style requirements, including those for file naming. The PLOS ONE style templates can be found at https://journals.plos.org/plosone/s/file?id=wjVg/PLOSOne_formatting_sample_main_body.pdf and https://journals.plos.org/plosone/s/file?id=ba62/PLOSOne_formatting_sample_title_authors_affiliations.pdf 2. PLOS requires an ORCID iD for the corresponding author in Editorial Manager on papers submitted after December 6th, 2016. Please ensure that you have an ORCID iD and that it is validated in Editorial Manager. To do this, go to ‘Update my Information’ (in the upper left-hand corner of the main menu), and click on the Fetch/Validate link next to the ORCID field. This will take you to the ORCID site and allow you to create a new iD or authenticate a pre-existing iD in Editorial Manager. 3. We note that you have included the phrase “data not shown” in your manuscript. Unfortunately, this does not meet our data sharing requirements. PLOS does not permit references to inaccessible data. We require that authors provide all relevant data within the paper, Supporting Information files, or in an acceptable, public repository. Please add a citation to support this phrase or upload the data that corresponds with these findings to a stable repository (such as Figshare or Dryad) and provide and URLs, DOIs, or accession numbers that may be used to access these data. Or, if the data are not a core part of the research being presented in your study, we ask that you remove the phrase that refers to these data. 4. Your ethics statement should only appear in the Methods section of your manuscript. If your ethics statement is written in any section besides the Methods, please move it to the Methods section and delete it from any other section. Please ensure that your ethics statement is included in your manuscript, as the ethics statement entered into the online submission form will not be published alongside your manuscript. 5. If the reviewer comments include a recommendation to cite specific previously published works, please review and evaluate these publications to determine whether they are relevant and should be cited. There is no requirement to cite these works unless the editor has indicated otherwise. 

Reviewers' comments:

**Comments to the Author**

1. Is the manuscript technically sound, and do the data support the conclusions?

Reviewer #1: No

Reviewer #2: Yes

2. Has the statistical analysis been performed appropriately and rigorously?

Reviewer #1: No

Reviewer #2: Yes

3. Have the authors made all data underlying the findings in their manuscript fully available?

Reviewer #1: Yes

Reviewer #2: Yes

4. Is the manuscript presented in an intelligible fashion and written in standard English?

Reviewer #1: No

Reviewer #2: Yes

**Reviewer #1:**  The authors Michal et al describe in their publication the observation how the DNase activity and ecDNA in plasma of patients with Testicular Germ Cell Tumors correlate with the clinical outcome. Furthermore, they investigated in a mouse model if a DNase application in combination with cisplatin treatment leads to a better outcome. This study is in general of interest, as it combines clinical findings and based on this new treatment approaches. However, the study needs some improvements as well as some more descriptions in the material and methods part and a clear structure of the manuscript and the figures.

Major:

The clear red line of the story is lost, because of the structure of the results part. A lot of data are presented in a very confusing way. For example in line 445 the header is “Comparison of the ecDNA in GCTs patients to HDs”, however presented are more or less all measured values like NE and MPO.

The samples for the study were collected over a long period and analysed several years later (2023). Even I understand that such studies are challenging in collecting enough samples, did the authors prove if the storage time influenced the data outcome and statically adjusted the results? Especially DNase activity can even if samples are stored at -80°C over time loose activity. In addition, how long was the storage time of the healthy controls? Could this explain the differences between both groups? The authors need to prove this influence by statistic analysis.

The study cohort contains 117 patients. Why did the authors only use 19 controls? This is less than 20% of the cohort. In addition, how is “healthy” defined? Non tumor patient?

Figure 1: It is unclear why the authors used an axis until 150 months even none of the data are ending in 150 month. In addition, some of the graphs end at 100 month others before. Was the observation time-frame of the patients not the same. Why are there missing data points?

Furthermore, it is unclear how the authors defined high and low values and therefore categorized the data. Furthermore, it is unclear how many patients belong now to which category? This should be clarified and the figure 1 needs to be presented as one clear figure.

It would be helpful to show a regression line of DNase activity versus DNA to present the described part as a figure.

Table 2: Why is not always the n of HD 19 in all measurements? Why is the total number (n) of GCT not always 117? And why is the number of patients in the different categories varying by measured parameters? The stage of the GCT should be always the same, or? In addition, the table header is again incorrect. The authors need to check all headers and the following content.

Table 4:

How can the n number between the four determined parameters differ in regard of high and low DNase? Either they are classified based on a mean as above and below and should never change or something is not correct presented. How do the authors explain this?

In addition, none of this values result in 117 patients. Why are some patients excluded in this assay?

Can the authors exclude that this effect in the assays is not the results of other parameters? How is the healthy donor group reacting in these assays? This would be a useful experiment.

The authors describe that they used plasma from a biobank, how were the PBMC isolated for the comet assay from biobank samples?

Animal Experiment:

The authors need to clarify how many animals per group / per experiment were used. It seems that it were more than one experiment? The authors describe a total of 16 mice in one experiment, however in Fig 3c in each group 8 points = 8 mice exist. This is confusing. How many mice are used in experiment 1 and 2?

Fig 4: Scale bars are missing. What can be seen in this pictures? Why are only pictures of with and without DNase are shown. The results text is not helpful. How was the result in combination of DNase and CDDP?

Is the tumor volume in Fig 4D statistically different?

In line 514 the authors describe data not shown. Nowadays all data should be shown in a publication.

Line 541 ff: The authors described they used plasma samples from a biobank. From where are the results coming of the blood cell counts? The blood count data should be shown. And in addition the data named in line 550 (data not shown). In routine blood diagnostic cell types like CD1c positive cells are not determined.

The supplemental data, especially in the graphs, needs to be revised, as they are now difficult to see, and everything should be aligned in the same direction for the reader. Which figure is shown on page 12? Numbering is missing. Figure legends should also be included under the Figure and not in the manuscript results included.

NETs:

The authors measured NE and MPO as NET markers in the patients. However, both could results from neutrophils undergoing apoptosis. Why did the authors not measure other NET markers, eg. H3cit, or MPO/Histone complexes? In addition, DNase can degrade NETs and NETs are described to be involved in metastasis processes. A NET quantification in the mice would be of high relevance to better understand the pathomechanism how DNase acts. In addition the discussion lacks the NET part and the

Minor:

Not all abbreviations used in the abstract are explained e.g. ncDNA, PFS and OS, which makes it complicated to understand everything. This needs to be explained.

The authors should proof read the whole manuscript for DNase and use always the correct writing, instead of switching to DNAse.

**Reviewer #2:**  The study addresses a clinical need to gain new biomarkers and therapeutic strategies for refractory testicular germ cell tumors. It is the first study to correlate plasma DNase activity with survival outcomes in TGCT patients. Translational approach combining clinical cohort analysis and preclinical in vitro/in vivo models is valuable.

The study includes relatively large patient cohort, comprehensive biomarker assessment and rigorous methodology.

It is written in a clear way with no major drawbacks.

Limitations:

Control group (n=19) is quite small compared to the patient cohort, limiting statistical robustness.

Survival data are promising but relatively short follow-up may underestimate long-term effects.

Impact beyond TGCTs (generalizability to other cancers) is implied but not tested here.

Recommendation

The dichotomization by mean value of ecDNA/DNase may oversimplify data distribution; more refined cut-offs would possibly improve prognostic accuracy.

The discussion should emphasize the preliminary nature of these preclinical data.

The limitations (small control group, single xenograft model, relatively short follow-up) should be more explicitly acknowledged.

**Do you want your identity to be public for this peer review?** For information about this choice, including consent withdrawal, please see our Privacy Policy

Reviewer #1: **Yes:** Nicole de Buhr

Reviewer #2: No

---

## [Author Response · Author response to Decision Letter 1]

7 Oct 2025

To: Dr. May R. Berenbaum

Editor-In-Chief

Plos One

Dear Dr. Berenbaum,

We would like to submit our revised manuscript entitled „Low Level of Plasma DNase Is Associated with Worse Clinical Outcome in Testicular Germ Cell Tumor Patients and Exogeneous DNase I Improves (Leverages) Cisplatin Treatment Efficacy” (PONE-D-25-29918) to your journal for publication. We would like to thank all reviewers for their time and effort. We have followed all their suggestions and tried to improve the manuscript accordingly. Here is our point-by-point response.

Reviewer #1:

The authors Michal et al describe in their publication the observation how the DNase activity and ecDNA in plasma of patients with Testicular Germ Cell Tumors correlate with the clinical outcome. Furthermore, they investigated in a mouse model if a DNase application in combination with cisplatin treatment leads to a better outcome. This study is in general of interest, as it combines clinical findings and based on this new treatment approaches. However, the study needs some improvements as well as some more descriptions in the material and methods part and a clear structure of the manuscript and the figures.

Major:

The clear red line of the story is lost, because of the structure of the results part. A lot of data are presented in a very confusing way. For example in line 445 the header is “Comparison of the ecDNA in GCTs patients to HDs”, however presented are more or less all measured values like NE and MPO.

Thank you very much for this point. We corrected the headers across the manuscript to better describe the presented type of analysis and results. We hope that the improved version of the manuscript is now clearer or at least less confusing.

The samples for the study were collected over a long period and analysed several years later (2023). Even I understand that such studies are challenging in collecting enough samples, did the authors prove if the storage time influenced the data outcome and statically adjusted the results? Especially DNase activity can even if samples are stored at -80°C over time loose activity. In addition, how long was the storage time of the healthy controls? Could this explain the differences between both groups? The authors need to prove this influence by statistic analysis.

Thank you very much for this point. We added this information to the manuscript.

Median samples storage: 1569 days (range: 961 – 3479)

Median healthy donors’ samples storage: 806 days (range: 577 – 911)

P < 0.0001

DNase activity but not ecDNA level decrease with increasing storage time. This could be a source of bias, but not for the survival analysis and not for ecDNA that seems to be stable over time. Of course, a detailed analysis of the effects of storage would be needed to prove this in a more precise manner.

DNase vs. days of sample storage: Spearman correlation = 0.3023, p = 0.0009

ecDNA vs. days of samples storage: Spearman correlation = 0.0874, p = 0.3996

When we adjust comparison of GCT patients vs. healthy donors for days of sample storage as co-variate, the difference remains significant.

Analysis of Variance Table

Source Sum of Mean Prob Power

Term DF Squares Square F-Ratio Level (Alpha=0,05)

Days of storage 1 1.087632 1.087632 7.29 0.007835* 0.764394

Group 1 1.020622 1.020622 6.84 0.009939* 0.737817

* Term significant at alpha = 0.05

The study cohort contains 117 patients. Why did the authors only use 19 controls? This is less than 20% of the cohort. In addition, how is “healthy” defined? Non tumor patient?

Thank you very much for this point. The sample size of our controls was based on number available samples from our biobank. We preferred to avoid adding “fresh” or “new” samples without any storage time. Healthy donors (N = 19) were age-matched men without history of testicular and/or other cancer who were recruited and consented according to the IRB-approved protocol. Healthy donors were recruited from staff working in co-operating institutions except men from departments directly involved in the study.

Figure 1: It is unclear why the authors used an axis until 150 months even none of the data are ending in 150 month. In addition, some of the graphs end at 100 month others before. Was the observation time-frame of the patients not the same. Why are there missing data points?

Thank you very much for this helpful comment. The median follow-up for all GCT patients was 22.6 months (range: 0.1–100.4 months), and for those still alive 25 months (1.9–100.4 months). The observation time varied across study groups, which is why we applied Kaplan–Meier estimates and compared groups using the log-rank test. We agree that extending the x-axis beyond the maximum follow-up may cause confusion, and we have adjusted the axis accordingly in the revised Figure 1. We added new Figure 1 to the manuscript.

Furthermore, it is unclear how the authors defined high and low values and therefore categorized the data. Furthermore, it is unclear how many patients belong now to which category? This should be clarified and the figure 1 needs to be presented as one clear figure.

Thank you very much for this point. As we stated in statistical section “Data about ecDNA concentrations in plasma including pellet microparticles and NETs-associated markers were dichotomized into high and low groups based on the ecDNA mean value of all samples. Plasma DNase activity was dichotomized by the same approach using mean value of all samples.”

We agree that current Figure 1 may cause confusion, and we changed it based on reviewer suggestion. We added new Figure 1 to the manuscript.

It would be helpful to show a regression line of DNase activity versus DNA to present the described part as a figure.

Thank you very much for this point. We added suggested graph to the manuscript.

Spearman correlation: -0.2021, p = 0.0495

Table 2: Why is not always the n of HD 19 in all measurements? Why is the total number (n) of GCT not always 117? And why is the number of patients in the different categories varying by measured parameters? The stage of the GCT should be always the same, or?

Thank you very much for this point. I agree with the reviewer, however in some categories we no measurement of particular parameter was available. We added this point into study limitations. The volume of some of the samples was too low for the analysis of some of the parameters while others are less volume consuming. Also, subtle hemolysis or lipemia might affect some parameters, but not others. That could explain varying number of data available for statistical analysis.

In addition, the table header is again incorrect. The authors need to check all headers and the following content.

Thank you very much for this point. We corrected the headers across the manuscript to better describe the presented type of analysis and results. We hope that the improved version of the manuscript is now clearer.

Table 4:

How can the n number between the four determined parameters differ in regard of high and low DNase? Either they are classified based on a mean as above and below and should never change or something is not correct presented. How do the authors explain this? In addition, none of this values result in 117 patients. Why are some patients excluded in this assay? Can the authors exclude that this effect in the assays is not the results of other parameters?

Thank you very much for this point. This is due to availability of the data (DNA damage and/or differential white blood count). We added this point into study limitations. Also, the variable available volume of samples for the analyses affects the number of data points.

How is the healthy donor group reacting in these assays? This would be a useful experiment.

Thank you very much for this point. For healthy donor group neither blood counts nor DNA damage assays were performed. We added this point into study limitations.

The authors describe that they used plasma from a biobank, how were the PBMC isolated for the comet assay from biobank samples?

Thank you very much for this point. Complete blood count with differential was obtained from medical records, however, this was not done for all patients, as some of them had only complete blood count without differential. Results of Comet assay was available from previous studies, that were already published (Oncotarget. 2016 Nov 15;7(46):75996-76005., Clin Genitourin Cancer. 2019 Oct;17(5):e1020-e1025., Mutat Res Genet Toxicol Environ Mutagen. 2020 Jun-Jul;854-855:503200). We added this clarification to the manuscript.

Animal Experiment:

The authors need to clarify how many animals per group / per experiment were used. It seems that it were more than one experiment? The authors describe a total of 16 mice in one experiment, however in Fig 3c in each group 8 points = 8 mice exist. This is confusing. How many mice are used in experiment 1 and 2?

Thank you very much for this point. There were 2 groups of animals used. Each group contained 16 animals. Each animal was bearing 2 xenografts. Each point represents one xenograft. Information was included in the method section and figure legends.

Fig 4: Scale bars are missing. What can be seen in this pictures? Why are only pictures of with and without DNase are shown. The results text is not helpful. How was the result in combination of DNase and CDDP?

Is the tumor volume in Fig 4D statistically different?

Thank you very much for this point. More detailed description of the images was included in the result section and in the figure legends. The result of the combination of CDDP and DNase is statistically significantly different as indicated in the figure 3C, where the quantification is shown in detail. Xenografts after CDDP and DNase combinatorial treatment were significantly smaller according to their endpoint weight taken at autopsy.

In line 514 the authors describe data not shown. Nowadays all data should be shown in a publication.

Thank you very much for this point. We added Supplementary Table 5 to show the mentioned data.

Line 541 ff: The authors described they used plasma samples from a biobank. From where are the results coming of the blood cell counts? The blood count data should be shown. And in addition the data named in line 550 (data not shown). In routine blood diagnostic cell types like CD1c positive cells are not determined.

Thank you very much for this point. Complete blood count with differential was obtained from medical records, however, this was not done for all patients, as some of them had only complete blood count without differential. Results of immune cell profiling was available from previous studies, that were already published. For the purpose of this manuscript we correlated these data for patients for whom these were available (Int J Mol Sci. 2021 Jul 31;22(15):8281., Front Oncol. 2022 Mar 11;12:858797.) We added this clarification to the manuscript.

The supplemental data, especially in the graphs, needs to be revised, as they are now difficult to see, and everything should be aligned in the same direction for the reader. Which figure is shown on page 12? Numbering is missing. Figure legends should also be included under the Figure and not in the manuscript results included.

Thank you very much for this point. We corrected Figure 1 and we included figure legends to the Figures 1-3 at the appropriate manuscript section.

NETs:

The authors measured NE and MPO as NET markers in the patients. However, both could results from neutrophils undergoing apoptosis. Why did the authors not measure other NET markers, eg. H3cit, or MPO/Histone complexes? In addition, DNase can degrade NETs and NETs are described to be involved in metastasis processes. A NET quantification in the mice would be of high relevance to better understand the pathomechanism how DNase acts. In addition the discussion lacks the NET part and the

Thank you very much for this point. Indeed, we agree with the reviewer that quantification of NETs would be ideal and needed. However, there is no commercially available assay for NETs quantification and there is a reason for it. None of the assays described in the literature are without technical or interpretational issues. NETs can be produced even without PAD activity and thus, citrullinated histones are likely not ideal. We have tried to establish the DNA/MPO sandwich ELISA and it worked, but not quantitatively – dilution experiments did not lead to expected outcomes. We, thus, do not want to publish results which are not backed up by technical verifications. NE and MPO are of course, also not ideal as NETs markers, but at least they are components of NETs and are present in the processed samples with verified and working ELISA assays.

Minor:

Not all abbreviations used in the abstract are explained e.g. ncDNA, PFS and OS, which makes it complicated to understand everything. This needs to be explained.

The authors should proof read the whole manuscript for DNase and use always the correct writing, instead of switching to DNAse.

Thank you very much for this point. We tried to correct these errors.

Reviewer #2:

The study addresses a clinical need to gain new biomarkers and therapeutic strategies for refractory testicular germ cell tumors. It is the first study to correlate plasma DNase activity with survival outcomes in TGCT patients. Translational approach combining clinical cohort analysis and preclinical in vitro/in vivo models is valuable.

The study includes relatively large patient cohort, comprehensive biomarker assessment and rigorous methodology.

It is written in a clear way with no major drawbacks.

Limitations:

Control group (n=19) is quite small compared to the patient cohort, limiting statistical robustness.

Thank you very much for this point. The sample size of our controls was based on number available samples from our biobank. We preferred to avoid adding “fresh” or “new” samples without any storage time. Healthy donors (N = 19) were age-matched men without history of testicular and/or other cancer who were recruited and consented according to the IRB-approved protocol. Healthy donors were recruited from staff working in co-operating institutions except men from departments directly involved in the study.

Survival data are promising but relatively short follow-up may underestimate long-term effects. Impact beyond TGCTs (generalizability to other cancers) is implied but not tested here.

Recommendation

The dichotomization by mean value of ecDNA/DNase may oversimplify data distribution; more refined cut-offs would possibly improve prognostic accuracy.

Thank you very much for this point. Thank you very much for this insightful comment. We agree that dichotomization by the mean value may not fully capture the complexity of the data distribution. However, we chose this approach for two reasons: (i) it provided a simple and transparent way of stratifying patients in this exploratory setting, and (ii) it allowed consistency and comparability with previously published studies that applied similar cut-offs. While more refined thresholds may potentially improve prognostic accuracy, they would also increase the risk of overfitting in our dataset of limited size. We therefore decided to keep the mean-based dichotomization in the current analysis, while acknowledging its limitations in the Discussion.

The discussion should emphasize the preliminary nature of these preclinical data.

The limitations (small control group, single xenograft model, relatively short follow-up) should be more explicitly acknowledged.

Thank you very much for this point. We agree with the reviewer and we added this point to the study limitations.

Moreover, we corrected the authors first and last names

We believe that our results could lead to better understanding the treatment resistance in GCTs.

We believe that your journal meets the expectation we address here.

Thank you in advance.

Kind regards,

Prof. Michal Mego, M.D., D.Sc.,

Head of 2nd Department of Oncology

Faculty of Medicine, Comenius University,

National Cancer Institute

Klenova 1,

---

## [Decision Letter · Decision Letter 1]

21 Oct 2025

Low Level of Plasma DNase Is Associated with Worse Clinical Outcome in Testicular Germ Cell Tumor Patients and Exogeneous DNase I Improves (Leverages) Cisplatin Treatment Efficacy

PONE-D-25-29918R1

Dear Dr. Michal,

We’re pleased to inform you that your manuscript has been judged scientifically suitable for publication and will be formally accepted for publication once it meets all outstanding technical requirements.

Kind regards,

Milad Khorasani, PhD

Academic Editor

PLOS ONE

Additional Editor Comments (optional):

Reviewers' comments:

Reviewer's Responses to Questions

**Comments to the Author**

Reviewer #2: All comments have been addressed

Reviewer #3: All comments have been addressed

2. Is the manuscript technically sound, and do the data support the conclusions?

Reviewer #2: Yes

Reviewer #3: Yes

3. Has the statistical analysis been performed appropriately and rigorously?

Reviewer #2: Yes

Reviewer #3: Yes

4. Have the authors made all data underlying the findings in their manuscript fully available?

Reviewer #2: Yes

Reviewer #3: Yes

5. Is the manuscript presented in an intelligible fashion and written in standard English?

Reviewer #2: Yes

Reviewer #3: Yes

Reviewer #2: All comments were addressed. The authors explained all uncertainties. Manuscript is suitable for publication.

Reviewer #3: As a newly invited reviewer for the revised version of this manuscript, I have carefully examined the authors’ responses to the previous reviewers’ comments as well as the revised manuscript itself. It is evident that the authors have made substantial efforts to address all the concerns raised during the initial review. The revised version demonstrates significant improvements in terms of methodological clarity, data presentation, and discussion depth.

The study provides solid evidence supporting the association between plasma DNase levels and clinical outcomes in testicular germ cell tumor patients. The additional experimental data and improved statistical analyses have strengthened the conclusions. Moreover, the preclinical data on the synergistic effect of DNase I with cisplatin are compelling and provide translational value. The manuscript is clearly written, well-structured, and scientifically sound. The authors’ revisions have successfully resolved the prior issues and enhanced the overall quality of the paper. Therefore, I find the revised version acceptable for publication in its current form.

**Do you want your identity to be public for this peer review?** For information about this choice, including consent withdrawal, please see our Privacy Policy

Reviewer #2: No

Reviewer #3: No

---

## [Editor Report · Acceptance letter]

PONE-D-25-29918R1

PLOS ONE

Dear Dr. Michal,

I'm pleased to inform you that your manuscript has been deemed suitable for publication in PLOS ONE. Congratulations! Your manuscript is now being handed over to our production team.

Kind regards,

on behalf of

Dr. Milad Khorasani

Academic Editor

PLOS ONE